# Formation of a β-barrel membrane protein is catalyzed by the interior surface of the assembly machine protein BamA

James Lee[1,2], David Tomasek[1,2†], Thiago MA Santos[2†], Mary D May[2†], Ina Meuskens[2], Daniel Kahne[1,2]*

[1]Department of Molecular and Cellular Biology, Harvard University, Cambridge, United States; [2]Department of Chemistry and Chemical Biology, Harvard University, Cambridge, United States

**Abstract** The β-barrel assembly machine (Bam) complex in Gram-negative bacteria and its counterparts in mitochondria and chloroplasts fold and insert outer membrane β-barrel proteins. BamA, an essential component of the complex, is itself a β-barrel and is proposed to play a central role in assembling other barrel substrates. Here, we map the path of substrate insertion by the Bam complex using site-specific crosslinking to understand the molecular mechanisms that control β-barrel folding and release. We find that the C-terminal strand of the substrate is stably held by BamA and that the N-terminal strands of the substrate are assembled inside the BamA β-barrel. Importantly, we identify contacts between the assembling β-sheet and the BamA interior surface that determine the rate of substrate folding. Our results support a model in which the interior wall of BamA acts as a chaperone to catalyze β-barrel assembly.

*For correspondence:
kahne@chemistry.harvard.edu

†These authors contributed equally to this work

Competing interests: The authors declare that no competing interests exist.

## Introduction

A major challenge in cell biology is to understand how proteins are incorporated into membranes. Transmembrane proteins can be divided into two main classes: α-helical bundles and β-barrels. Membrane incorporation of proteins with α-helical transmembrane segments has been studied extensively and is accomplished by the Sec machine, which functions in the cytoplasmic membrane of bacteria and in the endoplasmic reticulum of eukaryotes (*Rapoport et al., 2017*). In an α-helix, the backbone hydrogen bonds are internally satisfied, allowing a single α-helix to exist stably in a membrane, given that the side chains are compatible. β-barrel transmembrane proteins, which comprise a variable number of anti-parallel β-strands wrapped into a cylinder (*Koebnik et al., 2000*), are found in the outer membranes of Gram-negative bacteria, mitochondria, and chloroplasts. These proteins are involved in a range of functions, including the creation of pores to allow passage of diverse molecules across the membrane. Their assembly is thus essential for maintaining the integrity of the cell envelope (*Cho et al., 2014*; *Ruiz et al., 2005*; *Voulhoux et al., 2003*; *Wu et al., 2005*). A β-stranded structure is thought to be stable in the membrane only as a completely folded and closed β-barrel in which the N- and C-terminal β-strands are joined via hydrogen bonding and the membrane-exposed exterior surface is hydrophobic. Despite their importance, the mechanism by which β-barrel transmembrane proteins are folded is not well understood.

In Gram-negative bacteria, the β-barrel assembly machine (Bam) complex accelerates the folding and membrane integration of β-barrel transmembrane proteins (*Hagan et al., 2010*; *Wu et al., 2005*). Analogous machines exist in the outer membranes of mitochondria (Sam50, for sorting and assembly machinery) (*Gentle et al., 2004*; *Paschen et al., 2003*; *Wiedemann et al., 2003*) and another is proposed to exist in the outer membranes of chloroplasts (OEP80, outer envelope protein) (*Töpel et al., 2012*). In *Escherichia coli,* the Bam complex is composed of five proteins

(*Sklar et al., 2007*; *Wu et al., 2005*). The core of the complex is BamA, an essential protein that belongs to the Omp85 superfamily of outer membrane proteins that function as protein translocation or assembly factors (*Gentle et al., 2005*) and is conserved across all Gram-negative bacteria (*Heinz and Lithgow, 2014*; *Webb et al., 2012*). BamA contains five N-terminal soluble periplasmic polypeptide transport-associated (POTRA) domains and a C-terminal β-barrel transmembrane domain. The POTRA domains act as a scaffold that mediates interaction with four lipoproteins (BamB, BamC, BamD, and BamE) (*Kim et al., 2007*). Together, the periplasmic components of the complex create a protein vestibule beneath the membrane (*Bakelar et al., 2016*; *Gu et al., 2016*; *Han et al., 2016*). Although all four lipoproteins contribute to efficient folding, only BamD is essential (*Malinverni et al., 2006*) and found in all Gram-negative bacteria (*Heinz and Lithgow, 2014*; *Webb et al., 2012*). The Bam complex accelerates folding of β-barrels containing vastly different amino acid sequences and numbers of β-strands (*Doerner and Sousa, 2017*; *Hagan et al., 2010*; *Hagan and Kahne, 2011*; *Iadanza et al., 2016*; *Plummer and Fleming, 2015*; *Roman-Hernandez et al., 2014*). Therefore, this machine must accelerate folding by exploiting features common to its diverse substrates.

The prevailing model for folding is based on structures showing an open seam in the BamA barrel where the N- and C-terminal strands interact. It has been suggested that β-hairpins in the substrate assemble at the seam in what has been described as the 'budding model' because the nascent substrate barrel grows into the membrane as new strands are added at the seam (*Höhr et al., 2018*; *Noinaj et al., 2014*). A budding model has also been proposed for the mitochondrial ortholog of BamA called Sam50. It has been demonstrated that peptide substrate fragments crosslink strongly to the N-terminus of the Sam50 barrel and more weakly to its C-terminus (*Höhr et al., 2018*). Based on these experiments and the known structures of BamA, it was concluded that new β-strands were being added at the seam in accordance with the 'budding model'. An alternative model holds that an extensive region of β-sheet assembles in the periplasm at the POTRA domains of BamA with one end of the sheet held by one end of the seam (*Doerner and Sousa, 2017*; *Schiffrin et al., 2017a*). These models have focused largely how folding is initiated with less attention paid to explaining how folding is completed and substrates released.

Here, we have studied the folding of a large β-barrel, LptD, and variants that fold more slowly. LptD, a component of the lipopolysaccharide transport machine (*Bos et al., 2004*; *Braun and Silhavy, 2002*; *Sampson et al., 1989*; *Wu et al., 2006*), is one of the two essential β-barrel proteins in *Escherichia coli,* the other being BamA itself. LptD contains 26 β-strands, and must fold around a globular lipoprotein, LptE, which acts as a plug within the β-barrel (*Chng et al., 2010*; *Dong et al., 2014*; *Freinkman et al., 2011*; *Qiao et al., 2014*). LptD is useful as a model substrate for the Bam complex because it folds much more slowly (several orders of magnitude) than smaller β-barrel proteins (*Chng et al., 2012*; *Ureta et al., 2007*) making the process of folding more accessible for study than for other substrates. Moreover, we have previously identified a variant of LptD lacking a 23-amino acid stretch within β-strand seven and extracellular loop four (LptD4213) that accumulates as a late stage folding intermediate that can complete folding (*Lee et al., 2016*). Here, we take advantage of these slow folding substrates and in vivo crosslinking to identify contacts between folding intermediates and the Bam complex. The major conclusion from these crosslinking experiments is that LptD, in the process of folding, forms extensive contacts with the concave interior wall of the BamA β-barrel.

Thus, in contrast to either the budding or periplasmic models for folding, our evidence indicates that folding is catalyzed in the interior of the BamA β-barrel. In agreement with earlier models we show that the substrate LptD is held at its C-terminus through a stable interaction with BamA. In our model, however, the β-hairpins do not form at the lateral gate but rather form inside the BamA β-barrel, generating an extensive β-sheet as folding proceeds from the C-terminus towards the N-terminus. Here we present evidence that release of the N-terminus from the interior wall of BamA ultimately allows substrate β-barrel closure and insertion into the membrane. Importantly, we show that changes to residues in the interior of BamA can accelerate folding, leading us to conclude that it serves as an active site that catalyzes folding. Our results establish a model for β-barrel assembly by the Bam complex where the catalyst for β-strand formation is the interior surface of the BamA barrel.

## Results

### The interior surface and lateral gate of the BamA β-barrel form a binding site for substrates

We first sought to identify regions of the Bam complex that interact with LptD substrates. Individual components of the Bam complex can interact with substrates (*Bennion et al., 2010*; *Gessmann et al., 2014*; *Hagan et al., 2013*; *Ieva et al., 2011*; *Ieva and Bernstein, 2009*; *Pavlova et al., 2013*; *Plummer and Fleming, 2015*; *Ricci et al., 2012*; *Schiffrin et al., 2017b*), but how these interactions facilitate folding is unknown. Because it is essential, is conserved across all Gram-negative bacteria, and is the only transmembrane component in the Bam complex, we decided to focus on the role of the BamA β-barrel. We substituted residues throughout the BamA β-barrel with the unnatural amino acid para-benzoyl phenylalanine (pBPA) (*Chin et al., 2002*) to capture interactions between BamA and the substrates LptD and LptD4213. LptD4213 stalls as a late-stage folding intermediate during its assembly (*Lee et al., 2016*). This substrate was chosen because we expected that the longer residence time of LptD4213 on the Bam complex would allow for more efficient crosslink formation. We introduced pBPA substitutions at the lateral gate of BamA, where the N- and C-termini of the β-barrel meet (*Figure 1A*), and within the sixth extracellular loop (L6) (*Figure 1B*) because both the lateral gate and L6 were previously proposed to be important in the mechanism of substrate assembly (*Bamert et al., 2017*; *Höhr et al., 2018*; *Leonard-Rivera and Misra, 2012*; *Noinaj et al., 2014*). After photocrosslinking, we purified pBPA-containing BamA variants and assessed the presence of higher molecular weight adducts, representing BamA-substrate crosslinks, by immunoblotting.

We identified three residues at the lateral gate of BamA that showed a UV-dependent crosslink to LptD4213, one in the N-terminal β-strand one (N427) and two in the C-terminal β-strand 18 (Q803 and F804) (*Figure 1A, D and E*, and *Figure 1—figure supplement 1*). Additionally, pBPA substitutions at two positions in L6 (S657 and N666) yielded crosslinks to LptD4213 (*Figure 1B, D and E*, and *Figure 1—figure supplement 1*). It is important to note that these two residues in L6 flank the VRGF motif, the most conserved sequence of residues across the entire Omp85 superfamily (*Delattre et al., 2010*; *Leonard-Rivera and Misra, 2012*). The crosslinks at the lateral gate and L6 provide the first demonstration that these regions in BamA interact directly with substrate, but are consistent with other findings suggesting that these regions in the Omp85 superfamily members contact substrates (*Bamert et al., 2017*; *Höhr et al., 2018*).

If β-barrel substrates come in close contact with L6, which resides in the interior of BamA near the lateral gate, then substrates might insert directly at the lateral gate of BamA as previously suggested or might first enter the interior of the BamA β-barrel. The BamA interior surface has not been previously probed for interactions with substrates so we introduced pBPA at 18 positions spanning this surface (*Figure 1C*). Of those, we identified nine residues across β-strands two, four, five, seven and ten that showed UV-dependent crosslinks to substrate (*Figure 1C–E* and *Figure 1—figure supplement 1*). These nine residues and all but one of the four residues described above have their side chains oriented towards the interior of the BamA β-barrel (*Figure 1D*). The exception, F804, resides at the C-terminal edge of the lateral gate where the N- and C-terminal ends of the BamA β-barrel interact. Together, the residues that directly interact with substrate form an extensive surface that includes a substantial portion of the interior wall of the BamA β-barrel (*Figure 1D and E*, and *Figure 1—figure supplement 1B*). Notably, several interacting residues, including G528, G530, and K610, are located directly opposite of the lateral gate (*Figure 1D and E*). This is an important observation because it is not consistent with the proposal that the β-strands of substrates are sequentially added to the nascent β-barrel via interactions that occur between β-strands 1 and 16 at the BamA lateral gate (*Höhr et al., 2018*; *Noinaj et al., 2014*). Because our results show that the LptD4213 substrate forms interactions with the interior surface of BamA far from the lateral gate, substrates may begin to fold inside the BamA β-barrel before exiting through the lateral gate of BamA into the membrane.

Next, we asked whether full-length native substrates make the same contacts to BamA as the stalled substrate (LptD4213) during their assembly. Under native expression conditions, we did not observe crosslinks from BamA to wild-type LptD (*Figure 1A–C*). However, because crosslinking efficiency depends on the residence time a substrate is bound, we speculated that the crosslinking intensity might simply be too low to observe by immunoblotting. Therefore, we selected three

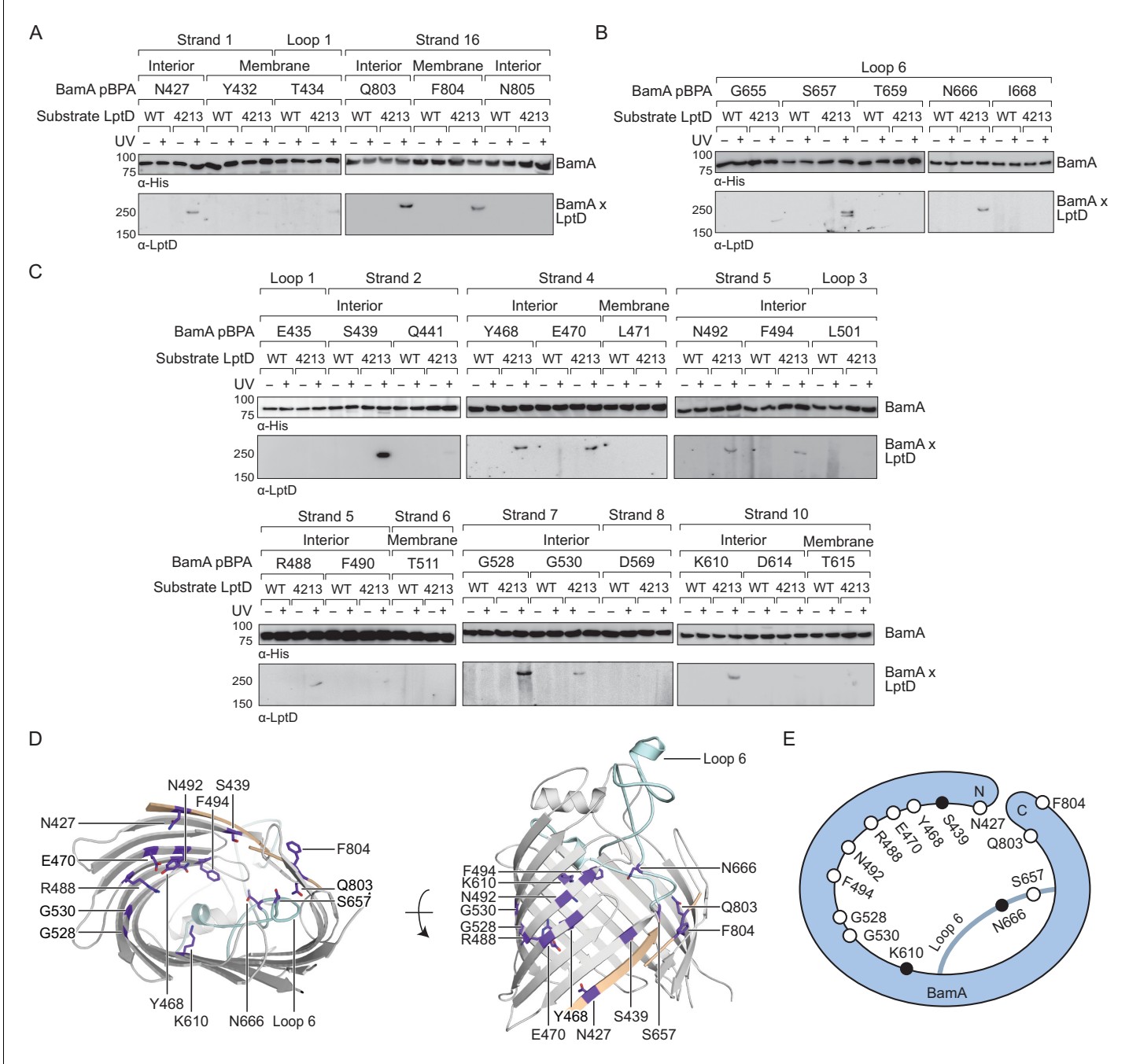

**Figure 1.** The interior wall and lateral gate of the BamA β-barrel form a substrate binding site. (A–C) Residues at the lateral gate (A), L6 (B), and interior wall (C) of BamA interact with substrate LptD during assembly. MC4100 and *lptD4213* (NR698) strains (expressing WT LptD or LptD4213, respectively) harboring the amber suppression system and expressing a His-tagged BamA (containing pBPA) were either left untreated or irradiated with UV light. Crosslinked adducts of BamA and substrate LptD/LptD4213 were identified by immunoblot analyses after Ni-NTA affinity purification. The orientation of the side chain of each residue in BamA substituted with pBPA is indicated (i.e., facing towards the membrane or interior of BamA). (D) Specific sites in the BamA β-barrel that interact with substrate LptD. Residues substituted with pBPA that crosslink to substrate are colored in purple. The first and last β-strands are colored in tan while the L6 loop is colored in cyan. Images were generated in PyMOL using the crystal structure of the BamA β-barrel from the *E. coli* BamABCDE complex (PDB: 5D0O). (E) Cartoon schematic of all sites in BamA that crosslink to substrate LptD. The view shown is the same as in the left panel of (D). Residues colored in black represent general substrate binding sites tested for crosslinking to full-length substrates (*Figure 2*). Additional views that include residues that do not form crosslinks are shown in *Figure 1—figure supplement 1*.

The online version of this article includes the following figure supplement(s) for figure 1:

**Figure supplement 1.** Crosslinking of the BamA interior to substrates.

representative sites at different locations within BamA (positions 439, 610, and 666 within the lateral gate, interior wall, and L6, respectively) that crosslinked to LptD4213. We assessed crosslinking to wild-type LptD that was expressed at higher levels (*Figure 2A*, *Figure 2—figure supplement 1*). Under these conditions, we observed UV-dependent crosslinks to the substrate at all three sites. Additionally, we observed crosslinks to other overexpressed smaller β-barrel substrates, including OmpF and LamB (*Figure 2B*, *Figure 2—figure supplement 1*). Therefore, the lateral gate, interior wall, and L6 of BamA, which all contact LptD4213 during its folding, also contact diverse native β-barrel substrates (*Figure 2C*). Our results imply that faster-folding native substrates follow a similar folding pathway as the slow-folding LptD4213.

## BamA and BamD bind non-overlapping regions within the C-terminal strands of substrates

The identification of a large surface of the BamA interior wall that interacts with substrates implies that a large portion of the stalled LptD4213 substrate interacts with BamA. We focused on defining contacts to BamA from residues in the substrate LptD barrel. It has been proposed that there exist

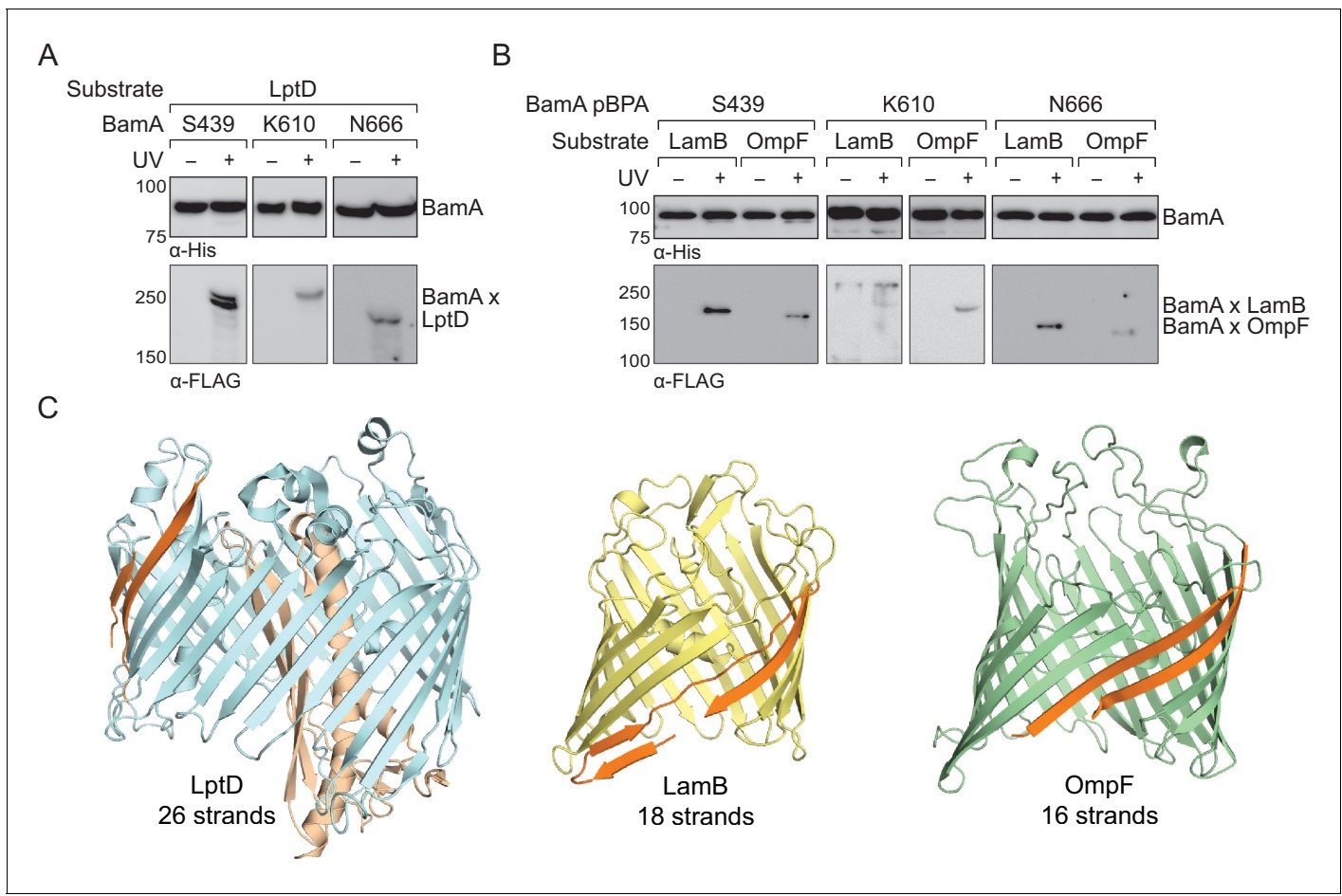

**Figure 2.** The interior of the BamA β-barrel forms a general substrate binding site. (**A–B**) The interior of the BamA β-barrel interacts with a diverse number of wild-type, full-length substrates including (**A**) LptD and (**B**) LamB and OmpF. MC4100 strains harboring the amber suppression system, expressing a His-tagged BamA (containing pBPA) and a FLAG-tagged substrate were either left untreated or irradiated with UV light. Crosslinked adducts of BamA to substrate were identified by immunoblot analyses after Ni-NTA affinity purification. (**C**) Structural diversity of substrates of the Bam complex. The first and last β-strands of each β-barrel are colored in orange. Images were generated using PyMOL from the structures of LptD/E (PDB: 4RHB), LamB (PDB: 1MAL), and OmpF (PDB: 2OMF). The assembly efficiency of these recombinant substrates are shown in *Figure 2—figure supplement 1*.

The online version of this article includes the following figure supplement(s) for figure 2:

**Figure supplement 1.** Analysis of cellular protein levels of recombinant outer membrane substrates.

recognition sequences in substrates that can interact with either BamA (*Robert et al., 2006*) or BamD during assembly (*Hagan et al., 2015*; *Lee et al., 2018*). Using pBPA substitutions near the C-terminus of LptD4213 (*Figure 3A–B*, *Figure 3—figure supplement 1*), we identified residues in β-strand 24 that interact with BamD, but not BamA (*Figure 3A and C*, *Figure 3—figure supplement 2A*) and residues in β-strands 25 and 26 that interact with BamA, but not BamD (*Figure 3B and C*, and *Figure 3—figure supplement 2B*). Even at native expression levels, several of these pBPA substitutions efficiently crosslinked wild-type LptD to BamD or BamA. These results are consistent with previous findings that the C-terminus of β-barrel substrates associate strongly with the Bam complex (*Hagan et al., 2015*; *Kutik et al., 2008*; *Lee et al., 2018*; *Robert et al., 2006*). The regions in the C-terminus of LptD that interact with BamA and BamD are spatially restricted and do not overlap, which would allow both components of the Bam complex to interact with the substrate simultaneously as has been previously suggested (*Ieva et al., 2011*).

## β-strand formation takes place in the interior of BamA

Crystal structures of wild-type LptD (*Dong et al., 2014*; *Qiao et al., 2014*) show that several residues in β-strands one and two contact the fourth extracellular loop (L4) of the protein. Deletion of this loop in LptD4213 (*Braun and Silhavy, 2002*; *Sampson et al., 1989*) causes this substrate to stall

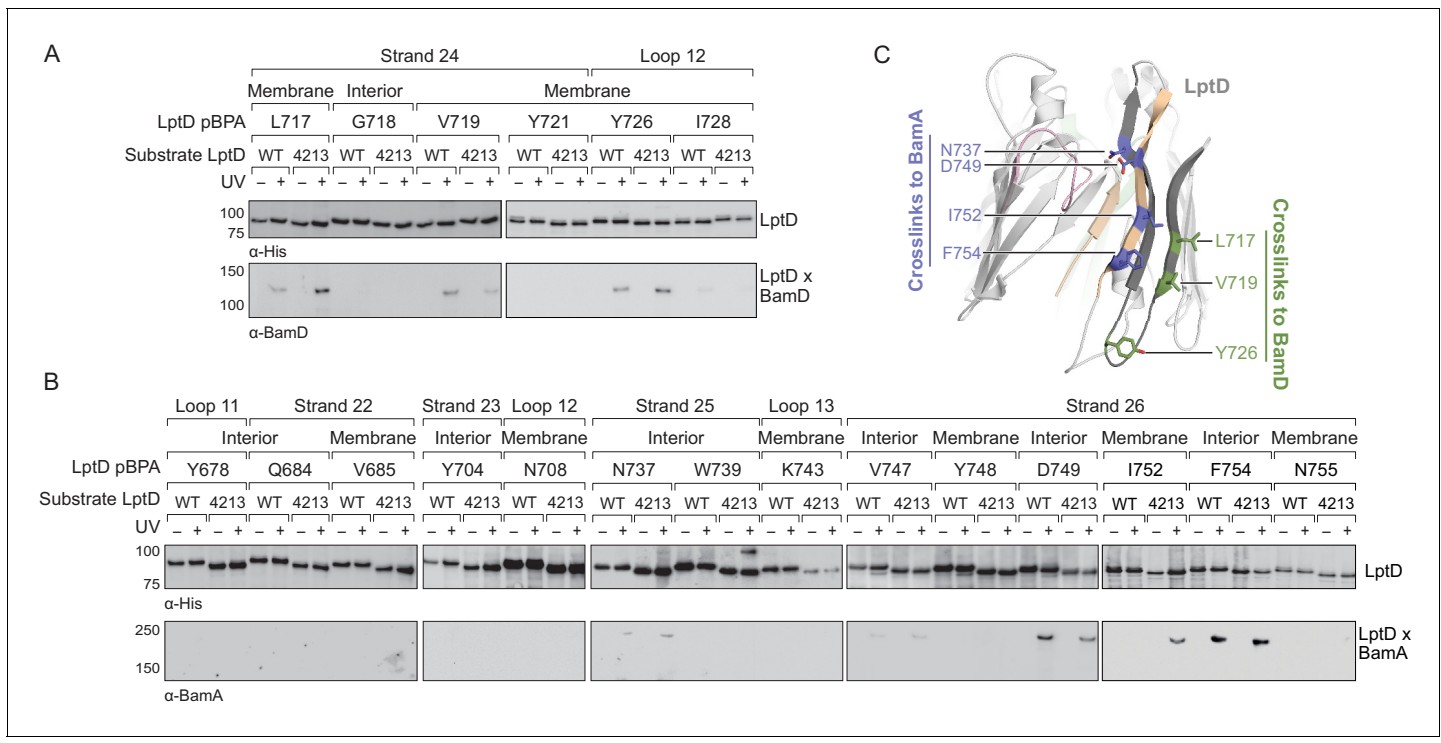

**Figure 3.** The C-terminal strands of LptD interacts with both BamA and BamD during assembly. (A) The third-to-last β-strand and final periplasmic loop of substrate LptD interact with BamD during assembly. MC4100 strains harboring both the amber suppression system and expressing a His-tagged LptD or LptD4213 (containing pBPA) were either left untreated or irradiated with UV light. Crosslinked adducts of BamD and substrate LptD/LptD4213 were identified by immunoblot analyses after Ni-NTA affinity purification. The orientation of the side chain of each residue substituted with pBPA is indicated (i.e., facing towards the membrane or interior of the folded form of LptD). (B) As in (A), but showing crosslinking to BamA from the final two β-strands and the final extracellular loop of substrate LptD. (C) Side view of LptD (gray) mapping the residues in the C-terminal strands that interact with the Bam complex. Residues in cyan and green form crosslinks to BamA and BamD, respectively. Extracellular loop 4 of LptD, part of the region deleted in LptD4213 (pink), interacts with the ends of the LptD β-barrel (tan). Images were generated in PyMOL using the crystal structure of *E. coli* LptD/E (PDB: 4RHB). Additional views that include residues that do not form crosslinks are shown in *Figure 3—figure supplement 1*. Additional crosslinking experiments demonstrating that the BamA and BamD substrate interaction sites do not overlap are shown in *Figure 3—figure supplement 2*.

The online version of this article includes the following figure supplement(s) for figure 3:

**Figure supplement 1.** Crosslinking of the C-terminal strands of substrate LptD to BamA and BamD.

**Figure supplement 2.** BamA and BamD bind non-overlapping regions within the C-terminal strands of substrate LptD.

on the Bam complex as an open β-barrel (*Lee et al., 2016*), implying that L4 is important for stabilizing closure. Regarding LptD4213, we found that residues in β-strands one and two, which would contact L4 in the folded state, instead formed crosslinks to BamA in a UV-dependent fashion (*Figure 4—figure supplement 1A*). Because BamA interacts with residues that would be involved in maintaining the closure of substrate once folded (*Figure 4—figure supplement 1B*), one function of BamA during assembly may be to bind substrates to prevent their premature release.

Next, we introduced pBPA substitutions farther away from the N-terminus of LptD. We tested 12 additional residues in β-strands four through six and the associated extracellular loops, which are N-terminal of the 23-residue deletion in LptD4213 (*Figure 4*, and *Figure 4—figure supplement 2*). The same substitutions were introduced into the wild-type substrate. Overall, the crosslinks to LptD4213 were more intense than for the wildtype substrate (*Figure 4A*), as expected given the

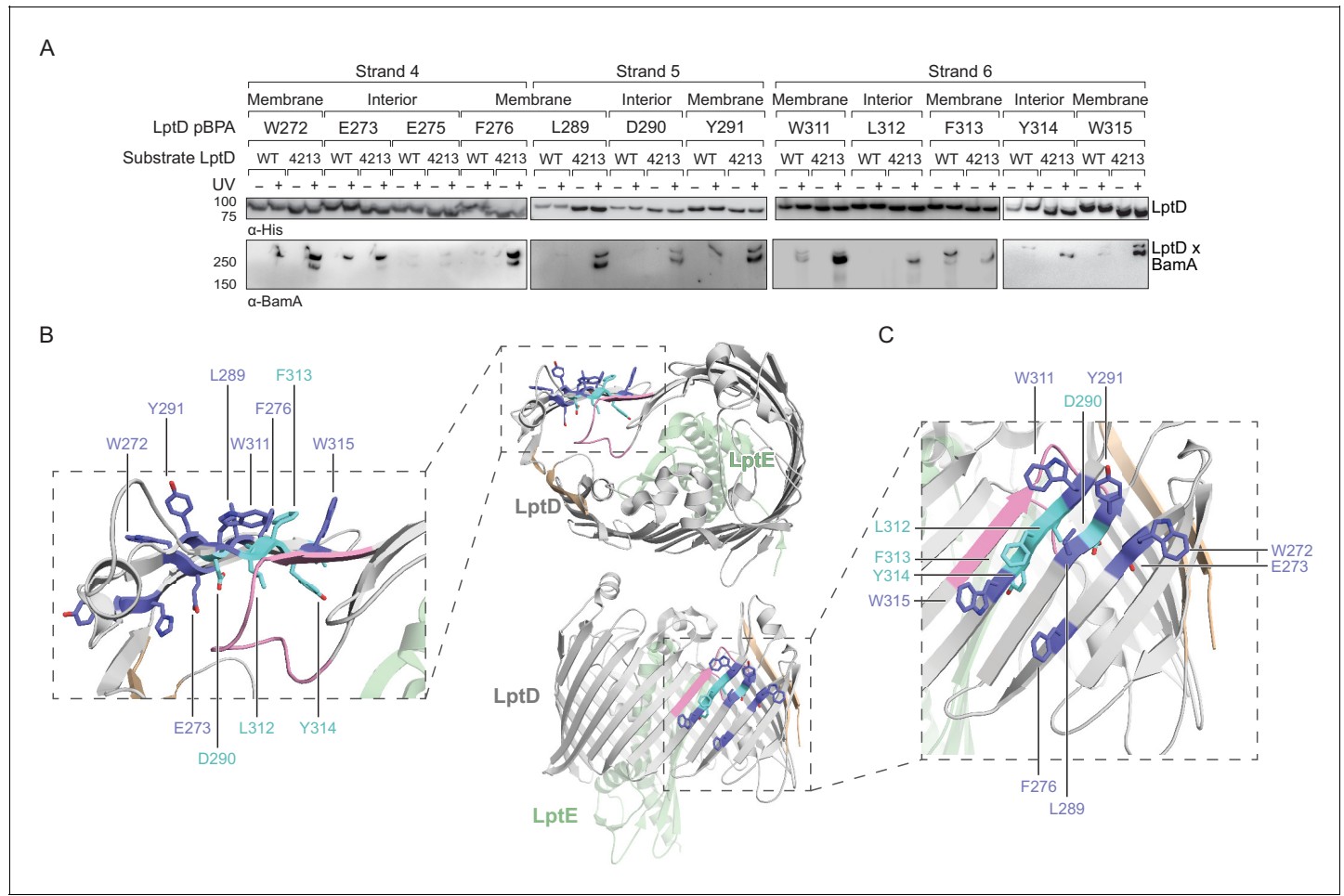

**Figure 4.** The N-terminal strands of LptD interacts with BamA during assembly. (**A**) β-strands four, five, and six of substrate LptD interact with BamA. Crosslinking was tested as described in *Figure 3*, but with pBPA substitutions in the N-terminal portion of substrate LptD/LptD4213. 'Membrane' and 'lumen' specify where the indicated residues would face in the mature barrel. (**B**) Top-down view of LptD showing that residues in at least 3 β-strands in the N-terminal region of the LptD barrel interact with BamA. Residues in LptD4213 that form strong crosslinks to BamA are shown in blue, while residues that form weak crosslinks are shown in cyan. The N- and C-terminal strands of LptD are indicated in tan, and LptE is shown in green. This color scheme is maintained in the rest of the figure. (**C**) Side view of LptD showing crosslinking positions as depicted in (**B**). Additional crosslinking experiments at residues in the first three strands of substrate LptD are shown in *Figure 4—figure supplement 1*. Additional views that include residues that do not form crosslinks are shown in *Figure 4—figure supplement 2*. Note that only crosslinks within a blot can be compared, and each blot includes only proximal residues.

The online version of this article includes the following figure supplement(s) for figure 4:

**Figure supplement 1.** Crosslinking of the first three strands of substrate LptD to BamA.
**Figure supplement 2.** Crosslinking of the N-terminal region of substrate LptD to BamA.

longer residence time of LptD4213 on Bam. We also noted that the crosslink intensities from several residues in β-strands four, five, and six of LptD4213 varied in a periodic pattern (*Figure 4A–C* and *Figure 4—figure supplement 2*). For example, in strand four crosslinking from residues W272 and F276 was more intense than from residue E275 (*Figure 4A*, left) and in strand five crosslinking intensities were greater for L289 and Y291 than for D290 (*Figure 4A*, middle). We did not observe periodicity in crosslinking intensities from strands one, two, and three (*Figure 4—figure supplement 1*). The periodicity or lack thereof implies that strands four, five and six have organized into a β-stranded structure but strands one, two, and three have not. Because we found an extensive surface within the interior of BamA that interacts with substrate (*Figure 1*), we propose that strands four, five, and six of the substrate are housed in the BamA barrel during the stall. Our data imply that BamA contains an extensive substrate binding site in its interior that can interact with substrates to chaperone β-sheet folding.

Our in vivo photocrosslinking data show that the C-terminus of the substrate (*Figure 3*) and an extensive region of the N-terminus of the substrate both interact with BamA (*Figure 4*). If this model is correct, it should be possible to form bidirectional crosslinks between the corresponding regions of BamA and the substrate. We substituted residues in BamA and LptD4213 with cysteines and probed for crosslink formation between the two proteins after treatment with 1,4-bis(maleimido) butane (BMB, 11 Å linker) and purification of crosslinked adducts formed by the two proteins. We found that a cysteine placed near the N-terminal strand of BamA (S439C) formed crosslinks to cysteines introduced near the C-terminal strand of LptD4213 (E733C and N737C, *Figure 5A and B*). Therefore, this establishes that C-terminus of substrate LptD is held at the N-terminus of the BamA barrel. Additionally, a cysteine placed in the L6 loop of BamA (N666), which resides within the interior of BamA, formed crosslinks to cysteines introduced near the N-terminus of LptD4213 (L245C, H262C and Y291C), but not to cysteines introduced near the C-terminus (*Figure 5A and B*). These

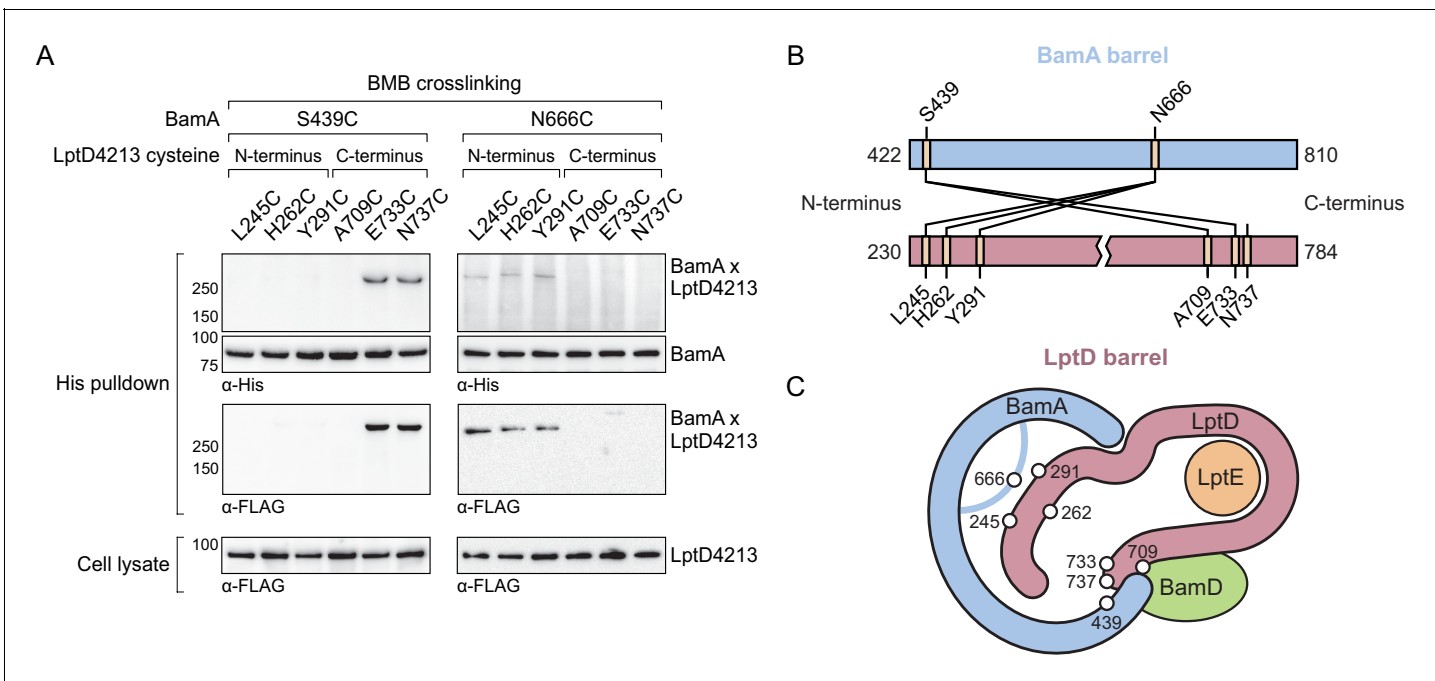

**Figure 5.** The N-terminal strands of the substrate are assembled within the BamA β-barrel. (**A**) The lateral gate and interior of the BamA β-barrel interacts with the ends of the LptD substrate. The N-terminal strands of BamA (S439) interacts with the C-terminal strands of substrate LptD, while L6 (N666), within the β-barrel of BamA, interacts with the N-terminal strands of substrate LptD. MC4100 strains expressing a His-tagged BamA cysteine mutant and a FLAG-tagged LptD4213 cysteine mutant were treated with the cysteine-cysteine crosslinker 1,4-bismaleimidobutane (BMB). Crosslinked adducts of BamA and substrate LptD4213 were identified by immunoblot analyses after Ni-NTA affinity purification. Immunoblots are provided that show expression levels of cysteine-containing LptD4213 constructs from total cell lysates (bottom). (**B**) Map of cysteine crosslinking between the BamA and LptD4213. (**C**) Top-down cartoon representation of LptD engagement by BamA/D based on cysteine crosslinking data.

results establish the topology of substrate LptD on the Bam complex (*Figure 5C*) and demonstrate that a large part of the N-terminus of the substrate is held within the interior of BamA.

## Substrate release from the interior wall of BamA facilitates β-barrel closure

Our data show that the interior wall of BamA interacts extensively with a large portion of the β-barrel substrate to chaperone folding. A model in which BamA catalyzes folding against its interior wall requires a rationale for how such an extensively bound substrate can be released once folding is complete. We and others have shown that no energy is required for folding on the Bam complex (*Doerner and Sousa, 2017*; *Hagan et al., 2010*; *Iadanza et al., 2016*; *Patel and Kleinschmidt, 2013*; *Plummer and Fleming, 2015*; *Roman-Hernandez et al., 2014*), and indeed, there is no ATP in the periplasm. The stalled complex provides a clue to the release mechanism. In fully-folded, wild-type LptD, L4 interacts with both the N- and C-termini of the β-barrel (*Figure 4—figure supplement 1*) (*Dong et al., 2014*; *Qiao et al., 2014*), suggesting that this loop stabilizes the released product relative to the late-stage intermediate that makes extensive contacts with the interior of BamA. When this loop is missing, as is the case in LptD4213, the interactions that normally stabilize the closed β-barrel are absent, and the relative energies of the BamA-associated intermediate and the fully closed β-barrel change such that the substrate stalls on the Bam complex at a late stage of folding.

This model predicts that mutations that destabilize the interactions between the N-terminus of LptD4213 and the interior of BamA may allow for release from the machine. An N274I amino acid substitution in LptD4213 was previously identified in a genetic selection for mutations that rescue permeability defects associated with LptD4213 (*Ruiz et al., 2005*). N274 is in β-strand four of the folded β-barrel (*Figure 6A*, left) with its polar side chain embedded within a nonpolar surface oriented towards the membrane (*Figure 6A*, right). Because our studies here identified crosslinks to BamA from β-strands flanking β-strand 4, we predicted that β-strand four itself would also interact with the interior wall of BamA. In β-strand 4, the side chains of residues 272 and 276 lie adjacent to that of residue 274 and are oriented towards the membrane (*Figure 6A*). Thus, pBPA substitutions at these positions should form crosslinks to BamA. To test this hypothesis, we replaced these residues with pBPA in both LptD and LptD4213 and probed for crosslinking to BamA (*Figure 6B*). Substitution at W272 and F276, two residues flanking N274, generated strong crosslinks to BamA in both wild-type LptD and the LptD4213 variant; weak crosslinks were also observed for both proteins after substitution of N274 with pBPA. These results suggest that β-strand 4 of LptD is bound to the interior wall of BamA during folding. Furthermore substitutions in β-strand four that disrupt polar contacts to the interior wall of BamA shorten the residence of the substrate at this binding site.

The weak crosslinks for the N274pBPA substitution suggested an explanation for how the N274I mutant suppresses the assembly defect of LptD4213. Because weaker crosslinking can indicate a shorter residence time, we speculated that the N274I suppressor weakens the affinity of the N-terminus of LptD for BamA. To test this possibility, we replaced residue 276, which is in the middle of β-strand four and faces outward, with pBPA in the LptD4213 N274I mutant. No crosslinking from this residue to BamA was observed (*Figure 6C*), consistent with a more transient interaction of β-strand four with the interior wall in the N274I variant. The N274I mutation evidently destabilizes the BamA-associated intermediate so that it is released more rapidly from the interior wall of BamA. Therefore, the ability of the N274I mutation to rescue folding is consistent with a release mechanism in which the intermediate is bound to BamA until folding has progressed to the point where the N- and C-termini are proximal. At this point, substrate β-barrel closure is promoted by interactions of the N- and C-terminal β-strands with loop 4, and the substrate detaches from the interior wall of BamA (*Figure 6D*).

## A slow folding LptD mutant is rescued by a compensatory mutation in the interior wall of BamA

We sought to test our model for BamA binding and release of substrates through rational design of slow-folding mutants and identification of compensatory mutations that rescue these folding defects. Because we have proposed that L4 plays a crucial role in stabilizing the folded substrate, we deleted a single amino acid at the start of loop 4 (D330) (*Figure 7—figure supplement 1A*) with the

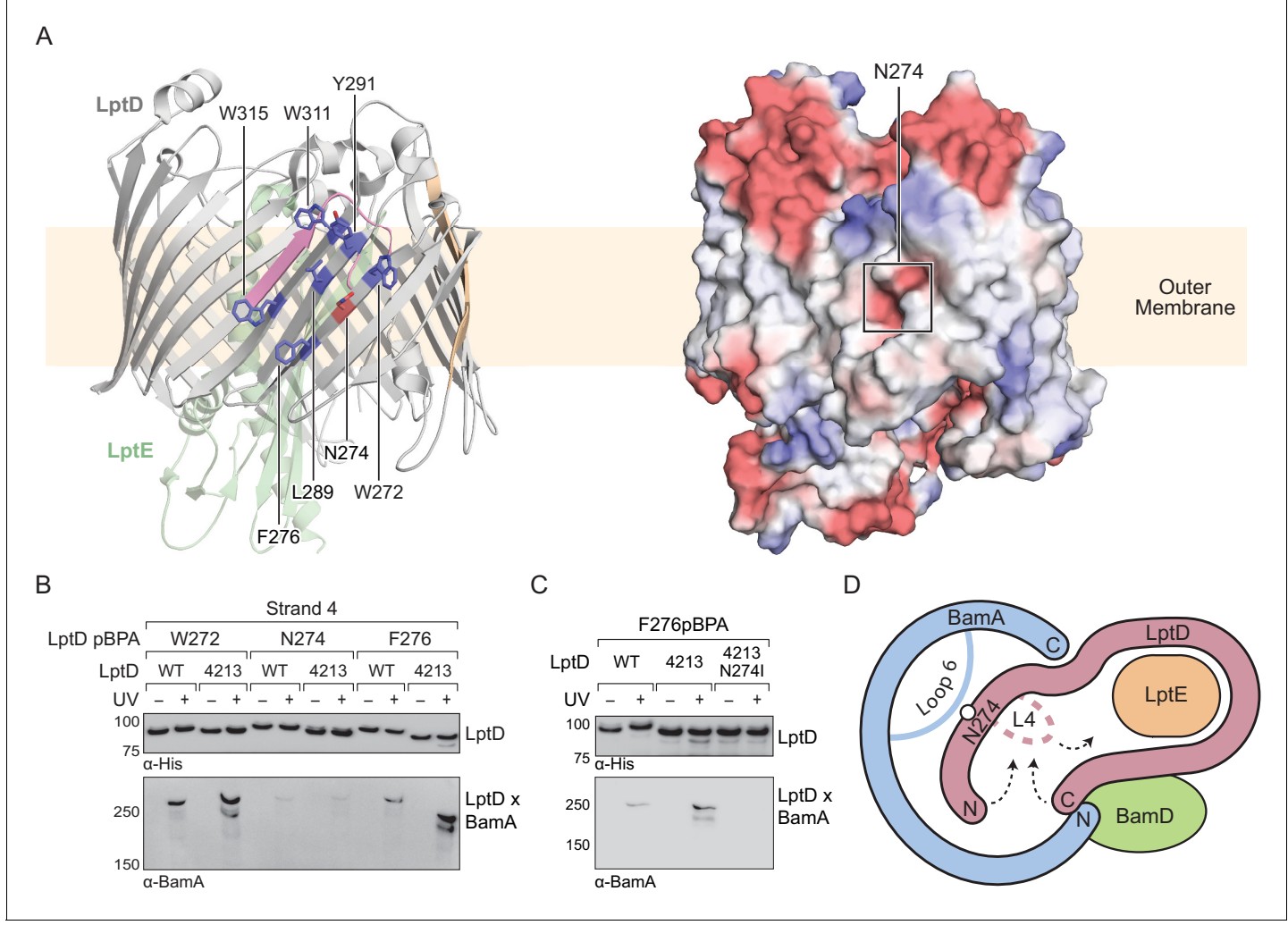

**Figure 6.** Substrate release from the interior wall of BamA allows β-barrel closure, triggering release from the Bam complex. (**A**) N274 resides within a large hydrophobic patch that encompasses at least six β-strands at the N-terminal region of the LptD β-barrel. The left panel shows the structure of LptD in cartoon form. The color scheme is the same as in **Figure 4**, with N274 indicated in red, and six hydrophobic residues that crosslink strongly to BamA in blue. The right panel shows an electrostatic surface plot generated using APBS, presented in the same orientation as the cartoon (left). Colors in the electrostatic surface plot represent potential rather than crosslinking residues. Red represents negative potential, white represents neutral potential, and blue represents positive potential. (**B**) The region around N274 directly interacts with BamA. Crosslinking was tested as described in **Figure 4**. (**C**) The N274I mutation suppresses the folding defects associated with LptD4213, allowing release from BamA, as judged by a reduction in crosslinking efficiency. (**D**) Model for substrate β-barrel closure and release from the Bam complex. N274I suppresses the folding defect associated with LptD4213 by facilitating release of the N-terminal strands of LptD from the interior of BamA.

expectation that this change would alter the disposition of the loop. Because the LptD$^{\Delta D330}$ substrate contains only a single amino acid deletion, it is more similar to wild-type LptD than the LptD4213 variant with its 23 amino acid deletion. Nevertheless, if the L4 loop is important for release because it stabilizes the folded β-barrel, then changing its orientation should affect folding.

We expressed either wild-type LptD, LptD4213, or LptD$^{\Delta D330}$ from a plasmid in otherwise wild-type *E. coli* and plated the strains on media with or without vancomycin to probe for changes in outer membrane integrity. Because LptD forms the translocon that delivers lipopolysaccharide to the cell surface, defects in LptD assembly result in outer membrane defects that allow the entry of antibiotics that otherwise are ineffective against *E. coli* (**Ruiz et al., 2005**; **Wu et al., 2006**). We found that expression of LptD$^{\Delta D330}$ increased susceptibility to vancomycin, consistent with impaired LptD assembly (**Figure 7A**, top three lanes). Since these strains also contain a second functional copy of LptD, the increased permeability conferred by LptD4213 and LptD$^{\Delta D330}$ is not result of a

lack of functional LptD translocons, but from a pore formed by the defective substrate. Next, we probed residence time of LptD$^{\Delta D330}$ on the Bam complex using crosslinking and observed that this variant formed strong crosslinks from the N-terminus of LptD$^{\Delta D330}$ to BamA, similar to LptD4213 and distinct from the wild-type LptD (*Figure 7B*, left panel). Therefore, LptD$^{\Delta D330}$, like LptD4213, has a longer residence time on BamA than wild-type LptD. Importantly, addition of the intragenic N274I substitution in LptD$^{\Delta D330}$ restored outer membrane barrier function (*Figure 7—figure supplement 1B*). Rescue of the assembly defect conferred by LptD$^{\Delta D330}$ is consistent with release of the N-terminus of LptD from BamA being the rate-limiting step in LptD$^{\Delta D330}$ assembly.

To assess the effect of the D330 deletion on the stability of the folded substrate, we analyzed the disulfide bond configuration of LptD and its variants (*Figure 7C*). LptD contains two disulfide bonds that, during its assembly, convert from an intermediate state containing consecutive disulfide bonds ([1,2] LptD) to the mature state containing nonconsecutive disulfide bonds ([1,3][2,4] LptD) (*Chng et al., 2012*; *Ruiz et al., 2010*). These two states can be detected by a difference in mobility by SDS-PAGE and serves as a proxy for folding status. In contrast to LptD4213, which largely remains in the intermediate disulfide bonded state, LptD$^{\Delta D330}$ eventually achieves the correctly folded state (*Figure 7C*, left half of immunoblot), indicating that it is on-pathway.

To evaluate the effect of the deletion of D330 on assembly kinetics, we pulse-labeled cells with [$^{35}$S]methionine and monitored the oxidation status of LptD over time (*Chng et al., 2012*). In cells expressing wild-type LptD, 50% of the substrate is converted to the mature disulfide bonded state within 20 min. In cells expressing LptD$^{\Delta D330}$, we found that the LptD$^{\Delta D330}$ substrate folded more slowly than WT LptD, as judged by a slower conversion to mature LptD with its native disulfide bond configuration ([1,3][2,4] LptD) (*Figure 7D*, top two panels).

Finally, we sought to identify additional mutations that could compensate for the folding defect in LptD$^{\Delta D330}$. We have shown that an intragenic suppressor that increases the hydrophobicity of the N-terminus of substrate LptD can influence the affinity of LptD to BamA and facilitate substrate release. If this is true, we predicted that mutations in the interior wall of BamA that weaken interactions with substrate would also facilitate substrate release. Therefore, we screened positions in the interior wall of BamA that we showed contact substrate (*Figure 7—figure supplement 2*) to identify a suppressor that would allow for more rapid release of the LptD$^{\Delta D330}$ substrate. We found that a single substitution in the interior of BamA, E470G, rescued the permeability barrier defects caused by LptD$^{\Delta D330}$ (*Figure 7A*, right bottom three lanes). Moreover, BamA$^{E470G}$ had a more transient association with LptD$^{\Delta D330}$ than with LptD4213 as judged by loss of crosslinking (*Figure 7B*, right panel). BamA$^{E470G}$ did not compromise the ability of LptD or the LptD$^{\Delta D330}$ mutant to achieve the mature disulfide bond configuration (i.e., properly oxidized product) (*Figure 7C*, right half of immunoblot). Finally, pulse-chase analysis showed that LptD$^{\Delta D330}$ was folded faster by BamA$^{E470G}$ ($t_{1/2}$=25 min) than by wild-type BamA ($t_{1/2}$=50 min), and its folding by BamA$^{E470G}$ was comparable to that of wild-type LptD by wild-type BamA (*Figure 7D*). The rate of folding of wild-type LptD was similar by wild-type BamA and by BamA$^{E470G}$. These results confirm that interactions between substrate and the interior wall of BamA are important for folding and that changes in these interactions affect an important step in folding, which is release from the machine.

## Discussion

Here, we present evidence for how the Bam complex catalyzes folding of a β-barrel substrate. The main features of the model are that the C-terminal strand of the substrate is held by BamA while the N-terminal strands of the substrate fold inside the BamA barrel. In our model the concave interior wall of BamA serves as an active site for formation of β-strands. At a late stage of assembly release of the completed β-sheet from the interior wall allows the substrate N-terminus to pair with the substrate C-terminus and close the β-barrel. Substrate β-barrel closure is further assisted by intramolecular interactions within the substrate, with substrate loop residues playing a critical role by interacting to bring both ends of the β-barrel together. In support of this model, we have identified a large substrate-binding surface that encompasses the interior wall of BamA. The periodicity in substrate residues that crosslink to BamA implies that folding intermediates have substantial β-sheet structure when bound to the interior surface of BamA. We also found that changes to residues in the substrate that contact the BamA interior or changes to residues in the BamA interior surface itself

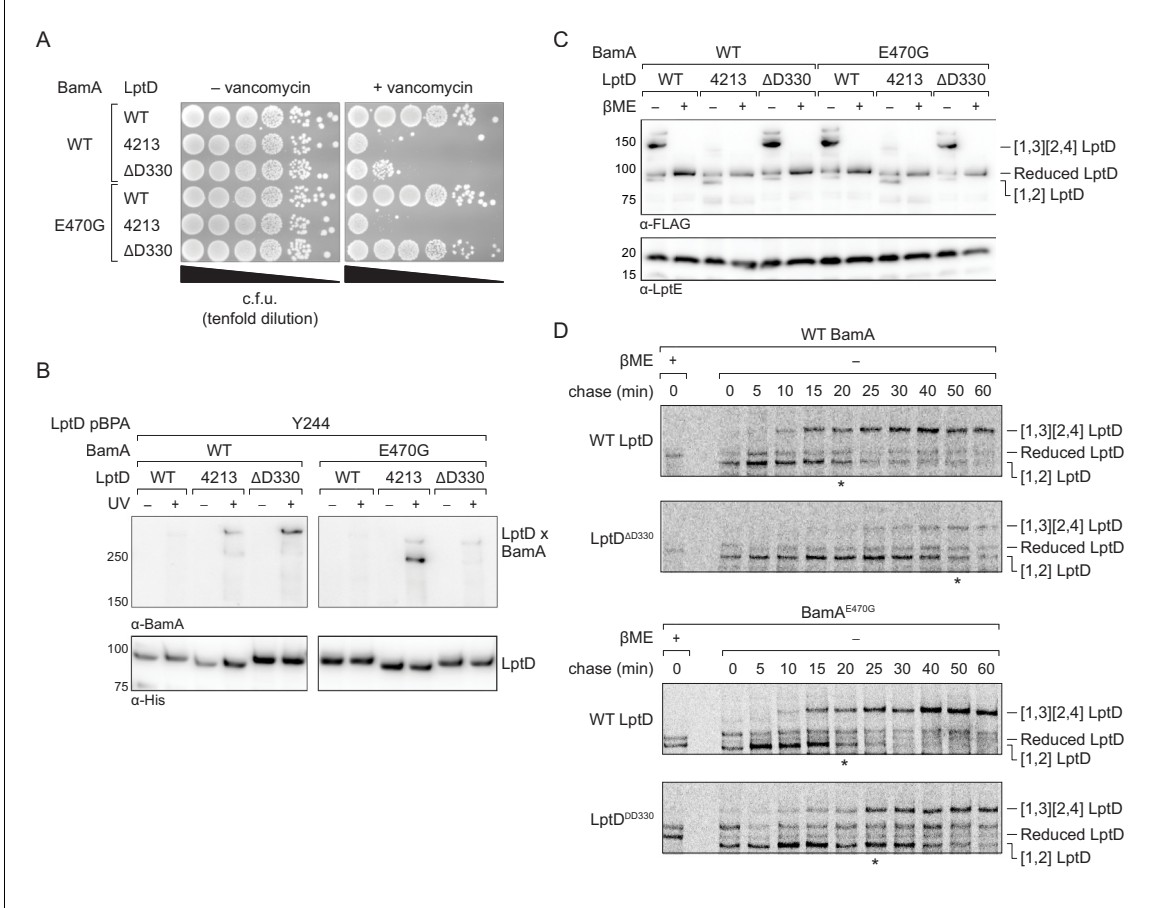

**Figure 7.** An assembly-defective LptD mutant is rescued by a compensatory mutation in the BamA interior wall. (**A**) Expression of LptD$^{\Delta D330}$ confers outer membrane permeability defects, while changes in BamA can suppress LptD$^{\Delta D330}$ associated-defects. MC4100 or *bamA*$^{E470G}$ cells were transformed with plasmids that express WT or mutant *lptD* alleles. Plating assays were performed on LB supplemented with 50 µg/mL vancomycin. (**B**) LptD$^{\Delta D330}$, like LptD4213, stalls on BamA during assembly. BamA$^{E470G}$ alleviates stalling of LptD$^{\Delta D330}$, as judged by a reduction in crosslinking efficiency, but does not alleviate stalling of LptD4213. (**C**) LptD$^{\Delta D330}$ can adopt the mature disulfide bonded state. MC4100 or *bamA*$^{E470G}$ cells expressing WT or mutant *lptD* alleles were harvested and analyzed via immunoblotting of cell lysates. When analyzed in the absence of reducing agent, LptD migrates at a molecular weight that reflects its state of assembly. Assembly of LptD involves the conversion of a reduced form to a form containing a disulfide bond between consecutive cysteines (designated [1,2]-LptD) that is then converted to the mature form, which contains disulfide bonds between nonconsecutive cysteines (designated [1,3][2,4]-LptD for the order in which the cysteines appear in the sequence). The [1,3][2,4] disulfide configuration reflects properly folded, functional LptD. (**D**) LptD$^{\Delta D330}$ is slow to mature into the functional disulfide bond configuration, while BamA$^{E470G}$ alleviates LptD$^{\Delta D330}$ assembly defects. MC4100 or *bamA*$^{E470G}$ cells expressing FLAG-tagged LptD(WT/ΔD330) were pulsed with [$^{35}$S] methionine and chased with cold methionine. Samples were subsequently immunoprecipitated using α-FLAG beads and analyzed by autoradiography. The asterisk below each autoradiograph represents the time point at which approximately 50% of the substrate has converted to the mature form (containing the [1,3][2,4] disulfide bond configuration). Other compensatory mutations that restore LptD$^{\Delta D330}$-associated defects are shown in *Figure 7—figure supplement 1* and *Figure 7—figure supplement 2*.

The online version of this article includes the following figure supplement(s) for figure 7:

**Figure supplement 1.** LptD$^{\Delta D330}$ partially phenocopies LptD4213-associated assembly defects.

**Figure supplement 2.** Compensatory mutations in BamA can rescue outer membrane permeability defects associated with stalled LptD substrates.

that contact substrate can increase or decrease the rate of folding. It follows that the interior surface of the BamA barrel wall is the catalyst for β-barrel strand formation.

An interesting aspect of our folding model is that the BamA interior surface is relatively polar (*Noinaj et al., 2013*), and yet it promotes the folding of β-barrel substrates with a hydrophobic exterior by forming extensive contacts with the hydrophobic surface of the growing substrate β-sheet. We argue that this type of association makes sense in the context of understanding the kinetics of β-strand formation. First, the polar nature of the interior wall is likely necessary because the interior

surface is exposed to water for at least some of the time and the cavity would not be stable if it was too nonpolar. Second, and more importantly, an extensively hydrophobic BamA interior surface would interfere with release of a hydrophobic β-sheet. That is, to ensure that substrate binding is not too strong as tight binding of a substrate would impede the rate of β-strand formation. Indeed, weak oriented binding of substrates is a hallmark of enzymatic catalysis. Third, release of bound water from the hydrophobic exterior surface of the substrate as it folds against the interior surface is likely the driving force for folding. Finally, the leading exposed edge of the β-sheet in the folding substrate contains unsatisfied hydrogen bonds and is therefore polar; a relatively polar interior may be required to stabilize this leading edge. Therefore, we propose that the apparent mismatch in polarity between the associating surfaces is in fact central to the enzymatic mechanism for how β-strand formation is catalyzed.

How then does the interior wall accelerate the folding of LptD and other substrates by nucleating the formation of β-strands against the BamA interior wall? The barrier to formation of β-sheet structure involves formation of an extended, entropically disfavored conformation of the peptide chain. However, once one β-hairpin forms, successive addition of more β-hairpins is facilitated in two ways. First, the leading edge of the β-sheet serves as a preorganized template facilitating the simultaneous formation of multiple hydrogen bonds. Second, each new β-strand has the appropriate hydrophobic periodicity of side chains to benefit from the hydrophobic effect as it packs against the interior wall. In short, confining substrates within a cage overcomes the entropic barrier that would normally slow folding because successive peptide β-strands do not have to sample as large a conformational space as they would in free solution, and moreover the new β-strands gain stability from their contacts to the cage itself.

Our model stands in contrast to the predominant model in the field of membrane β-barrel folding known as the budding model, which involves sequential insertion of individual β-hairpins at the seam of the BamA barrel (*Höhr et al., 2018*; *Noinaj et al., 2014*). This earlier model raises two important mechanistic problems. First, models in which hairpins are continuously added by forming hydrogen-bonding networks at the seam imply a substantial kinetic barrier for addition of each hairpin because the existing hydrogen-bonding networks must be disrupted to insert new hairpins. In contrast, in our model, the interior surface serves as a catalyst for β-strand formation without invoking continuous β-strand exchange with the machine. A second problem posed by the budding model is that, as the barrel grows, the N- and C-termini are displaced farther and farther apart, and this presents a problem for how barrel closure is achieved. No such problem arises in our model because confining the N-terminus of the growing sheet within the cage allows the ends of the substrate to remain close to each other as folding progresses.

Finally, we note that the mechanism that we have proposed for β-barrel folding by the Bam complex resembles the mechanism by which the GroEL/GroES chaperone system accelerates the folding of soluble proteins (*Hayer-Hartl et al., 2016*; *Horwich and Fenton, 2009*). The GroEL/GroES chaperonin uses ATP hydrolysis to drive a series of conformational changes that open and close the cavity to promote binding and release of folding intermediates (*Weissman et al., 1995*). Like GroEL/GroES, we propose that BamA uses a hydrophilic cage to limit the conformational space substrates can sample. Unlike GroEL/GroES, ATP is not required to drive conformational changes in BamA between its open and closed forms because the BamA β-barrel can access the open form without energy (*Bakelar et al., 2016*; *Gu et al., 2016*; *Iadanza et al., 2016*). Thus, the Bam complex accelerates the assembly of membrane β-barrel proteins by confining segments of folding substrates within the open BamA β-barrel to reduce the entropic cost of folding.

## Materials and methods

### Key resources table

| Reagent type (species) or resource | Designation | Source or reference | Identifiers | Additional information |
|---|---|---|---|---|
| Strain (*Escherichia coli*) | MC4100 | (*Casadaban, 1976*); PMID: 781293 | CGSC#: 6152 | See *Supplementary file 1* |

*Continued on next page*

Continued

| Reagent type (species) or resource | Designation | Source or reference | Identifiers | Additional information |
|---|---|---|---|---|
| Strain (*Escherichia coli*) | NR698 | (*Wu et al., 2005*); PMID: 15851030 | | See *Supplementary file 1* |
| Strain (*Escherichia coli*) | NR1134 | (*Lee et al., 2018*); PMID: 29463713 | | See *Supplementary file 1* |
| Strain (*Escherichia coli*) | JCM166 | (*Wu et al., 2005*); PMID: 15851030 | | See *Supplementary file 1* |
| Strain (*Escherichia coli*) | DEK1 | This paper | | See *Supplementary file 1* |
| Strain (*Escherichia coli*) | DH5α λ*pir* | (*Metcalf et al., 1994*; *Simon et al., 1983*) | | See *Supplementary file 1* |
| Antibody | Mouse monoclonal Anti-Penta His HRP conjugate | Qiagen | Cat#34460 | (1:5000) |
| Antibody | Mouse monoclonal Anti-FLAG M2 HRP conjugate | Sigma | Cat#A8592; RRID:AB_439702 | (1:50,000) |
| Antibody | Rabbit polyclonal Anti-BamA primary | (*Kim et al., 2007*); PMID: 17702946 | | (1:5000) |
| Antibody | Rabbit polyclonal Anti-BamD primary | (*Kim et al., 2007*); PMID: 17702946 | | (1:5000) |
| Antibody | Rabbit polyclonal Anti-LptD primary | (*Narita et al., 2013*); PMID: 24003122 | | (1:5000) |
| Antibody | Rabbit polyclonal Anti-LptE primary | (*Chng et al., 2010*); PMID: 22936569 | | (1:5000) |
| Antibody | Rabbit IgG, HRP-linked whole Ab (from donkey) | GE Healthcare | Cat#: NA935 | (1:5000) |
| Recombinant DNA reagent | Plasmids used | This paper | | See *Supplementary file 2* |
| Commercial assay | Amersham ECL Western Blotting Detection Kit | GE Healthcare | Cat#:RPN2232 | |
| Chemical compound | p-Benzoyl phenylalanine | Bachem | Cat#:4017646 | |
| Chemical compound | 10X Casein blocking buffer | Sigma | Cat#:B6429 | |
| Chemical compound | Anzergent 3–14 | Anatrace | Cat#:AZ314 | |
| Chemical compound | Ni-NTA Superflow | Qiagen | Cat#:30450 | |
| Chemical compound | BMB (1,4-bismaleimidobutane) | ThermoFisher | Cat#:22331 | |
| Chemical compound | L-cysteine hydrochloride | Alfa Aesar | Cat#:L06328 | |
| Chemical compound | TCEP-HCl | VWR | Cat#:97064 | |
| Chemical compound | S35 Methionine | American Radiolabeled Chemicals | Cat#:ARS 104A | |
| Chemical compound | N-ethylmaleimide | Sigma | Cat#:E3876 | |
| Chemical compound | Anti-FLAG M2 magnetic beads | Sigma | Cat#:M8823 | |
| Software | PyMol | Schrodinger | http://pymol.org; RRID:SCR_000305 | |

## Bacterial growth conditions

Unless otherwise noted, cultures were grown at 37°C and supplemented with the appropriate antibiotics and amino acids. Lysogeny broth (LB) and agar were prepared as described previously (*Silhavy et al., 1984*). When appropriate, carbenicillin (50 µg/ml), chloramphenicol (30 µg/ml), and kanamycin (50 µg/ml) were used. para-Benzoylphenylalanine (pBPA; Bachem Americas) was used at 0.9 mM.

## Strain construction

*E. coli* strains are presented in *Supplementary file 1*.

Cloning of the mutant gene *bamA*$^{E470G}$ into pDS132 was performed in *E. coli* DH5α λ*pir*. The resulting plasmid, pDS132::*bamA*$^{E470G}$, was purified and transformed into *E. coli* MC4100 for allelic exchange. Cells of the recipient strain were plated on LB agar supplemented with chloramphenicol to select transformants that integrated the plasmid into their chromosome. Following overnight growth at 37°C, one colony was inoculated into LB, incubated at 37°C for 4 hr, diluted in 1 × PBS, and plated on LB agar supplemented with 5% sucrose without NaCl. This step allowed selection of cells in which the integrated plasmid was excised from their chromosome. After overnight incubation at 37°C, about 50 colonies were streaked onto LB agar supplemented with chloramphenicol and on LB agar supplemented with 5% sucrose without NaCl. Clones that were resistant to sucrose and susceptible to chloramphenicol were screened by PCR and sequencing of the *bamA* locus.

## Plasmid construction

Plasmids are listed in *Supplementary file 2* and were constructed using traditional cloning methods and Gibson assembly. Vectors and insert DNA were generated by PCR with KOD DNA polymerase (Toyobo) and treated with *Dpn*I (NEB). Constructs were initially transformed into NovaBlue competent cells (Sigma) by heat shock. All plasmids were verified by Sanger sequencing.

## Site-specific in vivo photocrosslinking

Photocrosslinking experiments are based on techniques as previously described (*Freinkman et al., 2011*), with modifications. MC4100 strains harboring pSup-BpaRS-6TRN and pZS21*lptD-His* or pZS21*His-bamA* containing the TAG stop codon at the indicated positions were grown overnight, diluted 1:100 into 100 mL of the same media and grown to midlog phase. After normalization by optical density, each culture was split in half and directly irradiated with UV light at 365 nm for 10 min at room temperature. All samples were subsequently kept at 4°C. Samples were resuspended in 5 mL ice-cold TBS containing 1% Anzergent 3–14 (Anatrace), 100 µg/mL lysozyme, 1 mM PMSF, and 50 µg/mL DNase I, lysed by sonication, and centrifuged at 15,000 × *g* in a table-top centrifuge for 10 min. The supernatant was then passaged three times over Ni-NTA beads and then washed twice with 5 mL ice-cold TBS containing 0.02% Anzergent 3–14, and 20 mM imidazole. Samples were eluted with 1 mL ice-cold TBS containing 0.02% Anzergent 3–14 and 200 mM imidazole. Eluates were supplemented with 10% TCA by volume (100 µl) and incubated on ice for 30 min. Precipitated proteins were pelleted at 18,000 x g for 10 min at 4°C. All samples were resuspended in 50 µL of SDS-PAGE buffer and incubated at 95°C for 10 min. 8 µL of each sample were separated on 4–8% SDS-PAGE gels and analyzed by immunoblotting.

## In vivo BMOE chemical crosslinking

6 × His BamA variants containing the S439C or N666C mutation were cloned into the pZS21 vector. Substrates with a C-terminal 3 × FLAG tag (LptD4213 containing an N- or C-terminal cysteine) were cloned into the pTrc99a vector. MC4100 cells were transformed with one BamA-encoding plasmid and one substrate-encoding plasmid. The resulting strains were grown overnight in LB supplemented with 50 µg/mL carbenicillin, 50 µg/mL kanamycin, and 0.2% (w/v) glucose (37°C, 220 rpm). These overnight cultures were diluted 1:100 into 100 mL of fresh LB containing the same additives without glucose, and were grown (37°C, 220 rpm) to OD600 ∼ 0.5. Cells were then collected by centrifugation (4200 x g, 10 min, 4°C). Cell pellets were resuspended in PBS (20 mM NaH$_2$PO$_4$ pH 7.2, 150 mM NaCl). TCEP-HCl (VWR) was then added at a final concentration of 2 mM, and cells were incubated on a rocking platform (20 min, room temperature). Cells were then centrifuged (5000 x g,

10 min, 4˚C) and again resuspended in PBS. The cysteine-to-cysteine crosslinker 1,4-bis(maleimido) butane (BMB, Thermo Fisher Scientific) was added at a final concentration of 0.5 mM. After incubation on a rocking platform (40 min, room temperature), the crosslinking reaction was quenched via addition of L-cysteine hydrochloride monohydrate (Alfa Aesar) to a final concentration of 10 mM. Cells were centrifuged (5000 x g, 10 min, 4˚C) and the pellets were frozen at −80˚C prior to subsequent purification. In each sample, 6 × His BamA (and any associated substrate) was purified, and Ni-NTA elutions were subjected to SDS PAGE and subsequent Western blotting. 6 × His BamA was detected by using a penta-His (HRP) antibody (Qiagen). LptD4213−3 × FLAG (substrate) that was pulled down with 6 × His BamA was detected by using a monoclonal anti-FLAG M2-peroxidase (HRP) antibody (Sigma-Aldrich).

## Analysis of antibiotic sensitivities

Plating of the strains was performed as previously described (*Wzorek et al., 2017*). All strains were grown at 37˚C to an $OD_{600}$ of approximately 0.8. Cells were normalized to an $OD_{600}$ of 0.1 and then subject to five serial 10-fold dilutions. 5 µL of the dilution series were plated on agar plates containing the indicated additive and incubated at 37˚C for 18–20 hr.

## Analysis of cellular protein levels

All strains were grown at 37˚C to an $OD_{600}$ ~0.5. The cells from a 1 mL sample were normalized to an $OD_{600}$ of 0.3 and were collected by centrifugation at 10,000 x g for 10 min. The resulting cell pellets were resuspended in 100 µL of 1X SDS-sample buffer (+β-mercaptoethanol, β-ME) and incubated at 95˚C for 10 min. The samples were separated on 4–20% SDS-PAGE gels and analyzed via Immunoblotting. To analyze trimer assembly, samples were not boiled and were directly subject to SDS-PAGE.

## Immunoblotting

Proteins were transferred from Tris-glycine polyacrylamide gels to PVDF membranes (Bio-Rad) for 15 min at a constant voltage of 25 V. Membranes were blocked with casein blocking buffer (Sigma) for 1 hr and incubated with primary antibodies in blocking buffer at 4˚C overnight. Membranes were washed 3 times with TBST buffer (10 mM Tris·HCl pH 8.0, 150 mM NaCl, and 0.05% Tween-20), incubated with HRP-conjugated secondary antibodies for 1 hr at room temperature, and again washed 3 times with TBST buffer. Signal was detected using an Azure C400 imager (Azure Biosystems).

## Pulse-chase analysis

Pulse-chase experiments were performed as previously described (*Chng et al., 2012*). Briefly, a 5 mL culture was grown to an $OD_{600}$ of ~0.5 in M63 minimal media supplemented with 18 amino acids (minus methionine and cysteine) at 37˚C. The culture was pulse-labeled with [$^{35}$S]methionine (100 µCi/mL final concentration) (American Radiolabeled Chemicals) for 2 min and then chased with cold methionine (5 mM) at 37˚C. At the indicated time point during the chase, an 800 µL culture aliquot was transferred to a 1.5 mL tube containing 80 µL of TCA (70% in water) and incubated on ice for 20 min. Precipitated proteins were pelleted at 18,000 × g for 10 min at 4˚C, washed with 700 µL ice-cold acetone, and then solubilized in 80 µL 100 mM Tris·HCl, pH 8.0, containing 1% SDS and 20 mM N-ethylmaleimide (Sigma). The sample was sonicated for 30 s to aid solubilization. Following that, 800 µL of ice-cold immunoprecipitation (IP) buffer (50 mM Tris·HCl, pH 8.0, containing 150 mM NaCl, 2% Triton X-100, and 1 mM EDTA) was added and the sample was centrifuged at 18,000 × g for 10 min at 4˚C. 700 µL of the supernatant was transferred to another 1.5 mL tube containing 2.5 µL of anti-FLAG M2 magnetic beads 4 (Sigma). The beads were washed and preequilibrated with 3 × 1 mL IP buffer before use. The mixture was incubated on a rotary shaker for 1 hr at 4˚C, and the beads were washed three times with 800 µL of ice-cold high-salt buffer (50 mM Tris·HCl, pH 8.0, containing 1 M NaCl, 1% Triton X-100, and 1 mM EDTA) and one time with 800 µL ice- cold 10 mM Tris·HCl, pH 8.0, using a magnetic separation rack (New England Biolabs). Sixty microliters of 2 × SDS nonreducing sample buffer was then added to the beads and the mixture heated for 10 min at 100˚C to elute the bound proteins. Fifteen microliters of eluted sample was applied to SDS/PAGE directly. For reduction of disulfide bonds, 0.5 µL β-ME (Sigma) was added to 20 µL eluted sample and heated for 5 min at 100˚C before loading. Tris·HCl polyacrylamide gels (4–20%) were

used for SDS/PAGE analysis (running conditions: 150 V for 120 min). The gel was then dried and exposed to phosphor storage screens for autoradiography. Signals were detected with a Typhoon FLA 7000 image analyzer (GE Healthcare) using ImageQuant TL (GE Healthcare).

## Acknowledgements
We thank all members of the DK laboratory for helpful discussions. This work was supported by NIH grants F31GM116210 (to JL), T32AI132120 (to MDM), and AI081059 (to DK) and the HHMI Hanna H Gray Postdoctoral Fellowship to (TMAS).

## Additional information

### Funding

| Funder | Grant reference number | Author |
|---|---|---|
| National Institute of Allergy and Infectious Diseases | AI081059 | James Lee David Tomasek Thiago M A Santos Mary D May Ina Meuskens Daniel Kahne |
| National Institute of General Medical Sciences | F31GM116210 | James Lee |
| National Institute of Allergy and Infectious Diseases | T32AI132120 | Mary D May |
| Howard Hughes Medical Institute | Hanna H Gray Postdoctoral Fellowship | Thiago M A Santos |

The funders had no role in study design, data collection and interpretation, or the decision to submit the work for publication.

### Author contributions
James Lee, Conceptualization, Data curation, Investigation, Visualization, Methodology, Writing—original draft, Project administration, Writing—review and editing; David Tomasek, Thiago MA Santos, Mary D May, Data curation, Investigation, Visualization, Methodology, Writing—original draft, Project administration, Writing—review and editing; Ina Meuskens, Data curation, Investigation, Methodology, Writing—original draft, Project administration, Writing—review and editing; Daniel Kahne, Conceptualization, Supervision, Funding acquisition, Visualization, Writing—original draft, Project administration, Writing—review and editing

### Author ORCIDs
James Lee (iD) https://orcid.org/0000-0002-8551-0258
David Tomasek (iD) https://orcid.org/0000-0003-1212-4601
Thiago MA Santos (iD) https://orcid.org/0000-0002-0648-6078
Mary D May (iD) https://orcid.org/0000-0002-9484-8621
Ina Meuskens (iD) https://orcid.org/0000-0002-5103-1566
Daniel Kahne (iD) https://orcid.org/0000-0002-8296-1424

### Decision letter and Author response
Decision letter https://doi.org/10.7554/eLife.49787.sa1
Author response https://doi.org/10.7554/eLife.49787.sa2

## Additional files

### Supplementary files
• Supplementary file 1. List of strains used.

- Supplementary file 2. List of plasmids used.
- Transparent reporting form

## Data availability

All data generated or analyzed during this study are included in the manuscript and supporting files.

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
