## [Decision Letter]

**Acceptance summary:**

This study has used a biochemical strategy to investigate how new β-barrel proteins are folded by the β-barrel translocation apparatus formed by BamA. Using site-specific crosslinking from various side chains in BamA, the authors establish that folding of substrate occurs to a large extent in the interior of the BamA barrel. This conclusion is different than current models of how nascent β-barrel proteins fold, which had postulated that they 'bud' from BamA into the membrane. The new work will therefore warrant a re-assessment of how this important class of membrane proteins is made and folded, and will motivate new directions of study. The process of membrane protein insertion is a universally conserved process essential for life, and β-barrel proteins are found in many bacteria and the outer membranes of mitochondria and chloroplasts. For these reasons, the study will be of importance and interest to the readers of *eLife*.

**Decision letter after peer review:**

Thank you for submitting your article "Formation of a β-barrel membrane protein is catalyzed by the luminal surface of the assembly machine protein BamA" for consideration by *eLife*. Your article has been reviewed by three peer reviewers, and the evaluation has been overseen by Ramanujan Hegde as the Reviewing Editor and John Kuriyan as the Senior Editor. The reviewers have opted to remain anonymous.

The reviewers have discussed the reviews with one another and the Reviewing Editor has drafted this decision to help you prepare a revised submission.

All three referees found to paper to be important, well executed, and well written. No experimental revisions were judged to be absolutely necessary to support the central conclusions, although one referee asked whether the claim of non-overlapping interactions between substrate and different parts of the Bam complex might benefit from additional work. I leave the choice to add more experiments to buttress this claim to your discretion. I've appended the reviews below for your information. The most substantive points to be addressed have been extracted from the individual reviews and are re-iterated here for convenience.

1) The crosslinks that are observed are highly specific to mutant LptD unless the wild type is over-expressed at high levels. Can the authors briefly summarize in the Introduction the main reasons why the reader should treat this mutant as an on-pathway intermediate as opposed to an off-pathway product? This is of course critical for interpreting the crosslinks that are observed, and if the issue is not addressed at the outset, a reader will be wondering for some time whether they should believe the crosslinking results as assembly intermediates. Similarly, the authors should provide a statement about whether, under over-expression conditions, all of the protein does indeed mature. The overall concern here is that the crosslinks that are seen reflect off-pathway interactions.

2) Introduction: "Analogous machines exist in the outer membranes of mitochondria (Sam, for sorting and assembly machinery) (Paschen et al., 2003; Wiedemann et al., 2003) and chloroplasts (Toc, for translocon at the outer envelope membrane of chloroplasts) (Schnell et al., 1994)."

Consider citing Gentle et al. 2004 as well for the discovery of Sam50, it was called Omp85 in this original paper which also drew the link to bacterial Omp85 proteins.

The Omp85 component of the Toc, Toc75, is equivalent to a TpsB Omp85 and mediates the translocation of proteins through the plastid outer membrane. The likely counterpart to Sam50/BamA is a protein called Oep85 ((Töpel et al., 2012). The Schnell et al., 1994 reference is very old, much has happened in the field since then.

3) Introduction: "The Bam complex is composed of five proteins (Sklar et al., 2007; Wu et al., 2005)."

This statement could be modified to simply "In *E. coli*, the Bam complex is composed of five proteins (Sklar et al., 2007; Wu et al., 2005).". Alternatively, the additional information could be added to say that the BamA (Heinz et al., 2014) and BamD (Webb et al., 2012) subunits are found in all bacteria with outer membranes, while the other subunits evolved later and have a very restricted distribution (Webb et al., 2012). The advantage of this additional information is that it further highlights why the experiments in this paper have focused on features of the BAM complex concerning only BamA and BamD.

4) Figure 2: Showing the full panel for the western (Figure 2B) helps demonstrate the extremely high proportion of cross-linking (>50%) of LptD to BamA. The full western panel should also be shown for the BamD experiment (Figure 2A).

5) In Figure 1, Figure 2 and Figure 3, data regarding the residues tested are split between main and supplemental figures. This is confusing for the reader and puts results that are central to the study in the Supplementary files. The (A) panels from the Supplementary files – showing the crosslinking results – should be moved to the main text figures, and the structural models and cartoons in the main figures updated as needed. Similarly, the crosslinking results in Figure 2—figure supplement 2, panels (A) and (B) should be moved to the main text.

6) More information needs to be provided in the figures and legends to describe exactly what is being shown in the various crosslinking assays. In particular, it needs to be stated what protein is being used to pull down the complexes (the His-tagged protein in each case?) and what each blot is probing, i.e. cell lysates or the purified proteins. This is needed for the reader to understand the results. For example, in Figure 1, if the pulldown is on BamA-His and then the purified complexes are probed with anti-LptD, why is monomeric LptD present in each lane? Does LptD always form stable interactions with BamA? If this is true, then why aren't crosslinks being formed for the WT as well as 4231 LptD? Similarly, for experiments where pulldown is on LptD-His, why is BamA present in all lanes? Or LamB and OmpF in Figure 2—figure supplement 2?

7) In subsection “BamA and BamD bind non-overlapping regions within the C-terminus of substrates” and Figure 2, the authors present evidence for non-overlapping binding sites for BamA and BamD in the C-terminus of the LptD substrate. However, it is not apparent that the authors actually tested for overlapping binding. Thus, were strands 25 and 26 tested for binding to BamD and, conversely, was strand 24 tested for binding to BamA?

8) The authors state that their results show that β-strand formation in the substrate progresses from the C- to the N-terminus. However, it is not clear how the authors arrive at the directionality of folding from their data as presented. This needs to be explained or the text should be modified regarding this claim.

9) In subsection “β-strand formation takes place in the lumen of BamA and progresses from the C-terminus to N-terminus.” and Figure 3, the authors refer to a strong vs. weak crosslinks and a periodicity of strong crosslinks. There is concern here that the data are being over-interpreted, as the strength of crosslinking could be influenced by a number of different factors. Also, some type of quantitative analysis would need to be done to back up this claim, with appropriate normalization to protein expression levels, etc.

Reviewer #1:

This study uses a series of carefully designed site-specific crosslinking experiments combined with mutagenesis to probe the mechanism of β-barrel biogenesis mediated by the Bam complex. The key discovery is that the interior of the BamA barrel directly interacts with a nascent substrate, which is probably forming β-strands. The C-terminus of the substrate is held separately by other parts of the Bam complex. As more β-strands assemble, they progressively exit the lateral gate of the BamA barrel, until the final step when substrate releases from the BamA barrel interior. Cleverly designed mutagenesis within the substrate are used to investigate this key release step, with a mutant in BamA being identified that facilitates release. The findings are put together into a model of β-barrel assembly that makes energetic sense. This work is beautifully performed and the manuscript is written well. Of course, there are always many additional experiments one might ask for, but in my opinion, the findings make a convincing case that folding occurs on the interior surface of the BamA barrel and lead to a new model for β-barrel assembly.

Reviewer #2:

The process of outer membrane protein assembly is an essential aspect of bacterial cell biology. Integral membrane proteins with a β-barrel architecture are characteristic of all bacterial lineages with an outer membrane and, in species such as *E. coli*, these protein make up around half of the mass of the membrane. The β-barrel assembly machinery, including the core BAM complex, has been under investigation for several years but, despite impressive studies by several groups including the Kahne lab, the mechanism, by which the BAM complex functions, has been difficult to nail down.

The paper by Lee et al. addresses this issue by studying the details of the assembly of the relatively slow-assembling β-barrel protein LptD. The paper is clearly written and the problem at study is clearly defined for a general readership. The experiments are clearly described, and the data is of high quality. The conclusions are supported by the data and other relevant findings in the literature. I have just a few substantive concerns on the manuscript.

1) Introduction: "Analogous machines exist in the outer membranes of mitochondria (Sam, for sorting and assembly machinery) (Paschen et al., 2003; Wiedemann et al., 2003) and chloroplasts (Toc, for translocon at the outer envelope membrane of chloroplasts) (Schnell et al., 1994)."

Consider citing Gentle et al. 2004 as well for the discovery of Sam50, it was called Omp85 in this original paper which also drew the link to bacterial Omp85 proteins.

The Omp85 component of the Toc, Toc75, is equivalent to a TpsB Omp85 and mediates the translocation of proteins through the plastid outer membrane. The likely counterpart to Sam50/BamA is a protein called Oep85 ((Töpel et al., 2012). The Schnell et al., 1994 reference is very old, much has happened in the field since then.

2) Introduction: "The Bam complex is composed of five proteins (Sklar et al., 2007; Wu et al., 2005)."

This statement could be modified to simply "In *E. coli*, the Bam complex is composed of five proteins (Sklar et al., 2007; Wu et al., 2005).". Alternatively, the additional information could be added to say that the BamA (Heinz et al., 2014) and BamD (Webb et al., 2012) subunits are found in all bacteria with outer membranes, while the other subunits evolved later and have a very restricted distribution (Webb et al., 2012). The advantage of this additional information is that it further highlights why the experiments in this paper have focused on features of the BAM complex concerning only BamA and BamD.

3) Figure 2: Showing the full panel for the western (Figure 2B) helps demonstrate the extremely high proportion of cross-linking (>50%) of LptD to BamA. The full western panel should also be shown for the BamD experiment (Figure 2A).

Reviewer #3:

In their manuscript, Lee et al., use site-directed crosslinking to probe the mechanism of outer membrane protein (OMP) folding by the bacterial β-barrel assembly machine (Bam) complex. Great advances have been made over the past few years – including key contributions from this group – in determining how β-barrel OMPs are folded and inserted into the OM. This study addresses an important gap in knowledge, which is the precise mechanism by which the Bam complex facilitates folding of the substrate β-barrel and where this occurs on the central Bam component, BamA, which is itself a β-barrel OMP. Using a slow-folding mutant of the LptD substrate, the authors present convincing evidence for the formation of direct contacts between N- and C-terminal regions of the substrate and BamA, including regions inside the lumen of the BamA β-barrel. These results support a model – in contrast to the current budding or periplasmic models – in which substrates fold within the lumen of BamA. In the proposed model, the luminal surface of the BamA β-barrel acts as a chaperone for folding of the substrate β-barrel, which then releases from BamA upon forming a stable, closed structure. Overall, this is a well performed and presented study that adds important new information to the field. However, some concerns and questions regarding the data presentation and interpretation of results should be addressed, as detailed below.

1) In Figure 1, Figure 2 and Figure 3, data regarding the residues tested are split between main and supplemental figures. This is confusing for the reader and puts results that are central to the study in the Supplementary files. The (A) panels from the Supplementary files – showing the crosslinking results – should be moved to the main text figures, and the structural models and cartoons in the main figures updated as needed. Similarly, the crosslinking results in Figure 2—figure supplement 2, panels (A) and (B) should be moved to the main text.

2) More information needs to be provided in the figures and legends to describe exactly what is being shown in the various crosslinking assays. In particular, it needs to be stated what protein is being used to pull down the complexes (the His-tagged protein in each case?) and what each blot is probing, i.e. cell lysates or the purified proteins. This is needed for the reader to understand the results. For example, in Figure 1, if the pulldown is on BamA-His and then the purified complexes are probed with anti-LptD, why is monomeric LptD present in each lane? Does LptD always form stable interactions with BamA? If this is true, then why aren't crosslinks being formed for the WT as well as 4231 LptD? Similarly, for experiments where pulldown is on LptD-His, why is BamA present in all lanes? Or LamB and OmpF in Figure 2—figure supplement 2?

3) In subsection “BamA and BamD bind non-overlapping regions within the C-terminus of substrates” and Figure 2, the authors present evidence for non-overlapping binding sites for BamA and BamD in the C-terminus of the LptD substrate. However, it is not apparent that the authors actually tested for overlapping binding. Thus, were strands 25 and 26 tested for binding to BamD and, conversely, was strand 24 tested for binding to BamA?

4) The authors state that their results show that β-strand formation in the substrate progresses from the C- to the N-terminus. However, it is not clear how the authors arrive at the directionality of folding from their data as presented. This needs to be explained or the text should be modified regarding this claim.

5) In subsection “β-strand formation takes place in the lumen of BamA and progresses from the C-terminus to N-terminus.” and Figure 3, the authors refer to a strong vs. weak crosslinks and a periodicity of strong crosslinks. There is concern here that the data are being over-interpreted, as the strength of crosslinking could be influenced by a number of different factors. Also, some type of quantitative analysis would need to be done to back up this claim, with appropriate normalization to protein expression levels, etc.

---

## [Author Response]

All three referees found to paper to be important, well executed, and well written. No experimental revisions were judged to be absolutely necessary to support the central conclusions, although one referee asked whether the claim of non-overlapping interactions between substrate and different parts of the Bam complex might benefit from additional work. I leave the choice to add more experiments to buttress this claim to your discretion. I've appended the reviews below for your information. The most substantive points to be addressed have been extracted from the individual reviews and are re-iterated here for convenience.1) The crosslinks that are observed are highly specific to mutant LptD unless the wild type is over-expressed at high levels. Can the authors briefly summarize in the Introduction the main reasons why the reader should treat this mutant as an on-pathway intermediate as opposed to an off-pathway product? This is of course critical for interpreting the crosslinks that are observed, and if the issue is not addressed at the outset, a reader will be wondering for some time whether they should believe the crosslinking results as assembly intermediates. Similarly, the authors should provide a statement about whether, under over-expression conditions, all of the protein does indeed mature. The overall concern here is that the crosslinks that are seen reflect off-pathway interactions.

Cells expressing only LptD4213 are viable, which implies that at least some of this substrate can complete folding. LptD4213 folds slower than wild-type LptD and does accumulate on the Bam complex. To determine if the crosslinks we observed from the accumulated LptD4213 on the Bam complex are relevant to the wild-type folding pathway, we compared the crosslinking profiles between the wild-type LptD and LptD4213 and found similar crosslinks with wild-type LptD in many cases that were less intense than with LptD4213. Following the reviewer’s suggestion, we have now included two additional experiments in Figure 2—figure supplement 1 that establish that the wild-type substrate overexpression conditions used to observe crosslinks still leads to folded substrate.

2) Introduction: "Analogous machines exist in the outer membranes of mitochondria (Sam, for sorting and assembly machinery) (Paschen et al., 2003; Wiedemann et al., 2003) and chloroplasts (Toc, for translocon at the outer envelope membrane of chloroplasts) (Schnell et al., 1994)."Consider citing Gentle et al. 2004 as well for the discovery of Sam50, it was called Omp85 in this original paper which also drew the link to bacterial Omp85 proteins.The Omp85 component of the Toc, Toc75, is equivalent to a TpsB Omp85 and mediates the translocation of proteins through the plastid outer membrane. The likely counterpart to Sam50/BamA is a protein called Oep85 ((Töpel et al., 2012). The Schnell et al., 1994 reference is very old, much has happened in the field since then.

We have added the citations above (Introduction).

3) Introduction: "The Bam complex is composed of five proteins (Sklar et al., 2007; Wu et al., 2005)."This statement could be modified to simply "In E. coli, the Bam complex is composed of five proteins (Sklar et al., 2007; Wu et al., 2005).". Alternatively, the additional information could be added to say that the BamA (Heinz et al., 2014) and BamD (Webb et al., 2012) subunits are found in all bacteria with outer membranes, while the other subunits evolved later and have a very restricted distribution (Webb et al., 2012). The advantage of this additional information is that it further highlights why the experiments in this paper have focused on features of the BAM complex concerning only BamA and BamD.

We have modified the text as suggested (Introduction).

4) Figure 2: Showing the full panel for the western (Figure 2B) helps demonstrate the extremely high proportion of cross-linking (>50%) of LptD to BamA. The full western panel should also be shown for the BamD experiment (Figure 2A).

We have attached the full panels for the westerns (originally Figure 2, now part of Figure 3 and Figure 3—figure supplement 2).

5) In Figure 1, Figure 2 and Figure 3, data regarding the residues tested are split between main and supplemental figures. This is confusing for the reader and puts results that are central to the study in the Supplementary files. The (A) panels from the Supplementary files – showing the crosslinking results – should be moved to the main text figures, and the structural models and cartoons in the main figures updated as needed. Similarly, the crosslinking results in Figure 2—figure supplement 2, panels (A) and (B) should be moved to the main text.

We agree with the referee that this is a better organization for the data. All the crosslinking assays have now been incorporated into the main figures and Figure 1-figure supplement 1 has now been inserted as a new Figure 2.

6) More information needs to be provided in the figures and legends to describe exactly what is being shown in the various crosslinking assays. In particular, it needs to be stated what protein is being used to pull down the complexes (the His-tagged protein in each case?) and what each blot is probing, i.e. cell lysates or the purified proteins. This is needed for the reader to understand the results. For example, in Figure 1, if the pulldown is on BamA-His and then the purified complexes are probed with anti-LptD, why is monomeric LptD present in each lane? Does LptD always form stable interactions with BamA? If this is true, then why aren't crosslinks being formed for the WT as well as 4231 LptD? Similarly, for experiments where pulldown is on LptD-His, why is BamA present in all lanes? Or LamB and OmpF in Figure 2—figure supplement 2?

The Ni-NTA resin can nonspecifically bind membrane proteins, including BamA and LptD. The uncrosslinked LptD observed does not result from specific interactions with BamA (UV minus samples all have uncrosslinked LptD). LptD does not stably interact with BamA (previous work has shown that chemical crosslinking does not pull down wild-type LptD; Lee et al., 2016. We have modified the figure captions to better describe the experiments, and we have cropped immunoblots in the main figures to avoid confusion.

7) In subsection “BamA and BamD bind non-overlapping regions within the C-terminus of substrates” and Figure 2, the authors present evidence for non-overlapping binding sites for BamA and BamD in the C-terminus of the LptD substrate. However, it is not apparent that the authors actually tested for overlapping binding. Thus, were strands 25 and 26 tested for binding to BamD and, conversely, was strand 24 tested for binding to BamA?

The binding sites in BamA and BamD do not overlap. We have attached BamA and BamD immunoblots to demonstrate that we do not see crosslinking to BamA at the sites where we see crosslinking to BamD (new Figure 3—figure supplement 2).

8) The authors state that their results show that β-strand formation in the substrate progresses from the C- to the N-terminus. However, it is not clear how the authors arrive at the directionality of folding from their data as presented. This needs to be explained or the text should be modified regarding this claim.

We agree that we have not established the directionality of folding here. This is speculation based on the fact that the N-terminus of LptD4213 is being assembled within the interior of the BamA barrel while the C-terminus remains stably associated. We have removed this speculation from the text.

9) In subsection “β-strand formation takes place in the lumen of BamA and progresses from the C-terminus to N-terminus.” and Figure 3, the authors refer to a strong vs. weak crosslinks and a periodicity of strong crosslinks. There is concern here that the data are being over-interpreted, as the strength of crosslinking could be influenced by a number of different factors. Also, some type of quantitative analysis would need to be done to back up this claim, with appropriate normalization to protein expression levels, etc.

We have rewritten this section of text (subsection “β-strand formation takes place in the interior of BamA”) and modified the figures for clarity (originally Figure 3 and Figure 3—figure supplement 1, now Figure 4, Figure 4—figure supplement 1, and Figure 4—figure supplement 2). We agree with the reviewer about the general point that different factors can affect the intensity of crosslinking. We have now clarified that only proximal residues within a given blot are compared (see new Figure 4 caption). We consistently observe periodicity in crosslinking intensities for strands 4, 5, and 6. Relative intensities between blots cannot be compared, as we now state in the caption for Figure 4.

Reviewer #1:This study uses a series of carefully designed site-specific crosslinking experiments combined with mutagenesis to probe the mechanism of β-barrel biogenesis mediated by the Bam complex. The key discovery is that the interior of the BamA barrel directly interacts with a nascent substrate, which is probably forming β-strands. The C-terminus of the substrate is held separately by other parts of the Bam complex. As more β-strands assemble, they progressively exit the lateral gate of the BamA barrel, until the final step when substrate releases from the BamA barrel interior. Cleverly designed mutagenesis within the substrate are used to investigate this key release step, with a mutant in BamA being identified that facilitates release. The findings are put together into a model of β-barrel assembly that makes energetic sense. This work is beautifully performed and the manuscript is written well. Of course, there are always many additional experiments one might ask for, but in my opinion, the findings make a convincing case that folding occurs on the interior surface of the BamA barrel and lead to a new model for β-barrel assembly.Reviewer #2:The process of outer membrane protein assembly is an essential aspect of bacterial cell biology. Integral membrane proteins with a β-barrel architecture are characteristic of all bacterial lineages with an outer membrane and, in species such as E. coli, these protein make up around half of the mass of the membrane. The β-barrel assembly machinery, including the core BAM complex, has been under investigation for several years but, despite impressive studies by several groups including the Kahne lab, the mechanism, by which the BAM complex functions, has been difficult to nail down.The paper by Lee et al. addresses this issue by studying the details of the assembly of the relatively slow-assembling β-barrel protein LptD. The paper is clearly written and the problem at study is clearly defined for a general readership. The experiments are clearly described, and the data is of high quality. The conclusions are supported by the data and other relevant findings in the literature. I have just a few substantive concerns on the manuscript.1) Introduction: "Analogous machines exist in the outer membranes of mitochondria (Sam, for sorting and assembly machinery) (Paschen et al., 2003; Wiedemann et al., 2003) and chloroplasts (Toc, for translocon at the outer envelope membrane of chloroplasts) (Schnell et al., 1994)."Consider citing Gentle et al. 2004 as well for the discovery of Sam50, it was called Omp85 in this original paper which also drew the link to bacterial Omp85 proteins.The Omp85 component of the Toc, Toc75, is equivalent to a TpsB Omp85 and mediates the translocation of proteins through the plastid outer membrane. The likely counterpart to Sam50/BamA is a protein called Oep85 ((Töpel et al., 2012). The Schnell et al., 1994 reference is very old, much has happened in the field since then.

We thank the reviewer for this point and have incorporated these references as described above.

2) Introduction: "The Bam complex is composed of five proteins (Sklar et al., 2007; Wu et al., 2005)."This statement could be modified to simply "In E. coli, the Bam complex is composed of five proteins (Sklar et al., 2007; Wu et al., 2005).". Alternatively, the additional information could be added to say that the BamA (Heinz et al., 2014) and BamD (Webb et al., 2012) subunits are found in all bacteria with outer membranes, while the other subunits evolved later and have a very restricted distribution (Webb et al., 2012). The advantage of this additional information is that it further highlights why the experiments in this paper have focused on features of the BAM complex concerning only BamA and BamD.

We thank the reviewer for this point and have incorporated these references as described above.

3) Figure 2: Showing the full panel for the western (Figure 2B) helps demonstrate the extremely high proportion of cross-linking (>50%) of LptD to BamA. The full western panel should also be shown for the BamD experiment (Figure 2A).

Please see our reply to this point above.

Reviewer #3:In their manuscript, Lee et al., use site-directed crosslinking to probe the mechanism of outer membrane protein (OMP) folding by the bacterial β-barrel assembly machine (Bam) complex. Great advances have been made over the past few years – including key contributions from this group – in determining how β-barrel OMPs are folded and inserted into the OM. This study addresses an important gap in knowledge, which is the precise mechanism by which the Bam complex facilitates folding of the substrate β-barrel and where this occurs on the central Bam component, BamA, which is itself a β-barrel OMP. Using a slow-folding mutant of the LptD substrate, the authors present convincing evidence for the formation of direct contacts between N- and C-terminal regions of the substrate and BamA, including regions inside the lumen of the BamA β-barrel. These results support a model – in contrast to the current budding or periplasmic models – in which substrates fold within the lumen of BamA. In the proposed model, the luminal surface of the BamA β-barrel acts as a chaperone for folding of the substrate β-barrel, which then releases from BamA upon forming a stable, closed structure. Overall, this is a well performed and presented study that adds important new information to the field. However, some concerns and questions regarding the data presentation and interpretation of results should be addressed, as detailed below.1) In Figure 1, Figure 2 and Figure 3, data regarding the residues tested are split between main and supplemental figures. This is confusing for the reader and puts results that are central to the study in the Supplementary files. The (A) panels from the Supplementary files – showing the crosslinking results – should be moved to the main text figures, and the structural models and cartoons in the main figures updated as needed. Similarly, the crosslinking results in Figure 2—figure supplement 2, panels (A) and (B) should be moved to the main text.

Please see our reply to this point above.

2) More information needs to be provided in the figures and legends to describe exactly what is being shown in the various crosslinking assays. In particular, it needs to be stated what protein is being used to pull down the complexes (the His-tagged protein in each case?) and what each blot is probing, i.e. cell lysates or the purified proteins. This is needed for the reader to understand the results. For example, in Figure 1, if the pulldown is on BamA-His and then the purified complexes are probed with anti-LptD, why is monomeric LptD present in each lane? Does LptD always form stable interactions with BamA? If this is true, then why aren't crosslinks being formed for the WT as well as 4231 LptD? Similarly, for experiments where pulldown is on LptD-His, why is BamA present in all lanes? Or LamB and OmpF in Figure 2—figure supplement 2?

Please see our reply to this point above.

3) In subsection “BamA and BamD bind non-overlapping regions within the C-terminus of substrates” and Figure 2, the authors present evidence for non-overlapping binding sites for BamA and BamD in the C-terminus of the LptD substrate. However, it is not apparent that the authors actually tested for overlapping binding. Thus, were strands 25 and 26 tested for binding to BamD and, conversely, was strand 24 tested for binding to BamA?

Please see our reply to this point above.

4) The authors state that their results show that β-strand formation in the substrate progresses from the C- to the N-terminus. However, it is not clear how the authors arrive at the directionality of folding from their data as presented. This needs to be explained or the text should be modified regarding this claim.

Please see our reply to this point above.

5) In subsection “β-strand formation takes place in the lumen of BamA and progresses from the C-terminus to N-terminus.” and Figure 3, the authors refer to a strong vs. weak crosslinks and a periodicity of strong crosslinks. There is concern here that the data are being over-interpreted, as the strength of crosslinking could be influenced by a number of different factors. Also, some type of quantitative analysis would need to be done to back up this claim, with appropriate normalization to protein expression levels, etc.

Please see our reply to this point above.